# MiRNA-128 regulates the proliferation and neurogenesis of neural precursors by targeting PCM1 in the developing cortex

Wei Zhang[1†], Paul Jong Kim[2†], Zhongcan Chen[1], Hidayat Lokman[2], Lifeng Qiu[1], Ke Zhang[1], Steven George Rozen[3], Eng King Tan[4,5], Hyunsoo Shawn Je[2,5,6*], Li Zeng[1,5*]

[1]Neural Stem Cell Research Lab, Research Department, National Neuroscience Institute, Singapore, Singapore; [2]Molecular Neurophysiology Laboratory, Signature Program in Neuroscience and Behavioral Disorders, Duke-NUS Graduate Medical School, Singapore, Singapore; [3]Center for Computational Biology, Duke-NUS Graduate Medical School, Singapore, Singapore; [4]Department of Neurology, National Neuroscience Institute, Singapore, Singapore; [5]Neuroscience and Behavioral Disorders program, Duke-NUS Graduate Medical School, Singapore, Singapore; [6]Department of Physiology, Yong Loo Lin School of Medicine, National University of Singapore, Singapore, Singapore

**Abstract** During the development, tight regulation of the expansion of neural progenitor cells (NPCs) and their differentiation into neurons is crucial for normal cortical formation and function. In this study, we demonstrate that microRNA (miR)-128 regulates the proliferation and differentiation of NPCs by repressing pericentriolar material 1 (PCM1). Specifically, overexpression of miR-128 reduced NPC proliferation but promoted NPC differentiation into neurons both in vivo and in vitro. In contrast, the reduction of endogenous miR-128 elicited the opposite effects. Overexpression of miR-128 suppressed the translation of PCM1, and knockdown of endogenous PCM1 phenocopied the observed effects of miR-128 overexpression. Furthermore, concomitant overexpression of PCM1 and miR-128 in NPCs rescued the phenotype associated with miR-128 overexpression, enhancing neurogenesis but inhibiting proliferation, both in vitro and in utero. Taken together, these results demonstrate a novel mechanism by which miR-128 regulates the proliferation and differentiation of NPCs in the developing neocortex.

*For correspondence: shawn.je@
duke-nus.edu.sg (HSJ); Li_Zeng@
nni.com.sg (LZ)

†These authors contributed
equally to this work

**Competing interests:** The
authors declare that no
competing interests exist.

**Reviewing editor:** Eunjoon Kim,
Korea Advanced Institute of
Science and Technology,
Republic of Korea

## Introduction

Neurogenesis, the process by which functionally integrated neurons are generated from neural progenitor cells (NPCs), involves the proliferation and neuronal fate specification of NPCs and the subsequent maturation and functional integration of the neuronal progeny into neuronal circuits (*Gupta et al., 2002*). Given its importance in the development of the nervous system, neurogenesis is tightly regulated at many levels by both extrinsic and intrinsic factors (*Heng et al., 2010*), and its disruption has been associated with various pathologies, including autism spectrum disorders (ASDs), Treacher Collins syndrome, and various neural tube defects (*Sun and Hevner, 2014*). Therefore, uncovering the molecular mechanisms that underlie neurogenesis is crucial to understand the functions and plasticity of brain development and to prevent such pathologies (*Sun and Hevner, 2014*).

MicroRNAs (miRNAs) are small noncoding RNA molecules that function in the transcriptional and post-transcriptional regulation of gene expression in a variety of organisms (*Kawahara et al., 2012*).

**eLife digest** The neurons that transmit information around the brain develop from cells called neural progenitor cells. These cells can either divide to form more progenitor cells or to become specific types of neurons. If these carefully regulated processes go wrong – for example, if progenitors fail to stop dividing in order to mature – a range of neurodevelopmental conditions may develop, including autism spectrum disorders.

Small RNA molecules called microRNAs control gene activity and protein formation by targeting certain other RNA molecules for destruction. One such microRNA, called miR-128, helps newly formed neurons to move to the correct region of the cortex – the outer layer of the brain, which is essential for many cognitive processes including thought and language. However, it was not clear whether miR-128 plays any other roles in the development of neurons.

Zhang, Kim et al. have now analysed the role of miR-128 in the developing cortex of mice. The findings suggest that miR-128 prevents cortical neural progenitor cells from dividing and supports their development into more specialized cells. Causing miR-128 to be over-produced in the progenitor cells caused the cells to divide less often and encouraged them to mature into neurons. Conversely, removing miR-128 from the progenitor cells caused them to divide more and resulted in fewer neurons forming.

Further investigation revealed that miR-128 works by causing less of a protein called PCM1 to be produced. Without this protein, cells cannot divide properly. Future studies could now investigate in more detail how miR-128 and PCM1 affect how the neurons in the cortex develop and work.

Encoded by eukaryotic nuclear DNA, miRNAs function through imperfect base-pairing with complementary sequences in target mRNA molecules, typically triggering their targeted degradation or translational repression by the RNA-induced silencing complex RISC (*Bartel, 2009*; *Shi et al., 2010*). The role of miRNAs in neuronal development and function has recently received increased attention, and the specific spatiotemporal expression of these molecules may be essential for brain morphogenesis and neurogenesis (*Volvert et al., 2012*; *Zhang et al., 2014*). However, the specific miRNAs that regulate the proliferation and differentiation of NPCs during early cortical development are not well established.

In this study, we demonstrate that miR-128, which has previously been shown to play a crucial role in cortical migration (*Franzoni et al., 2015*), inhibits self-renewal and promotes neuronal differentiation in mouse NPCs by targeting pericentriolar material 1 (PCM1), a critical protein for cell division, during early cortical development (*Ge et al., 2010*). Specifically, ectopic overexpression of miR-128 reduces the proliferation of NPCs but promotes the differentiation of NPCs into neurons both in vivo and in vitro. Conversely, knockdown of miR-128 enhances proliferation but inhibits neuronal differentiation. Knockdown of endogenous PCM1 mimics the cellular phenotype of miR-128-overexpressing NPCs. Furthermore, the concomitant overexpression of both PCM1 and miR-128 in NPCs rescues the cellular phenotype associated with the overexpression of miR-128 in NPCs, indicating that PCM1 lies downstream of miR-128 and regulates the proliferation and neural fate specification of NPCs in vitro and in utero.

Taken together, our data indicate that miR-128 is an important regulator of neurogenesis in the embryonic cortex and suggest that aberrant miR-128 expression may account for the abnormal cortical development that underlies the pathophysiology of certain neuropsychiatric disorders, including autism.

## Results

### miR-128 is expressed in NPCs in the developing murine cortex

To determine the spatial distribution of miR-128 in the developing embryonic cortex, we performed in situ hybridization (ISH) using digoxigenin (DIG)-labeled locked nucleic acid (LNA) detection probes targeted to the mature form of miR-128 (*Figure 1A*). As previously reported (*Tan et al., 2013*), miR-128 was found to be predominantly expressed in the brains and spinal cords of E14.5 mice, and no

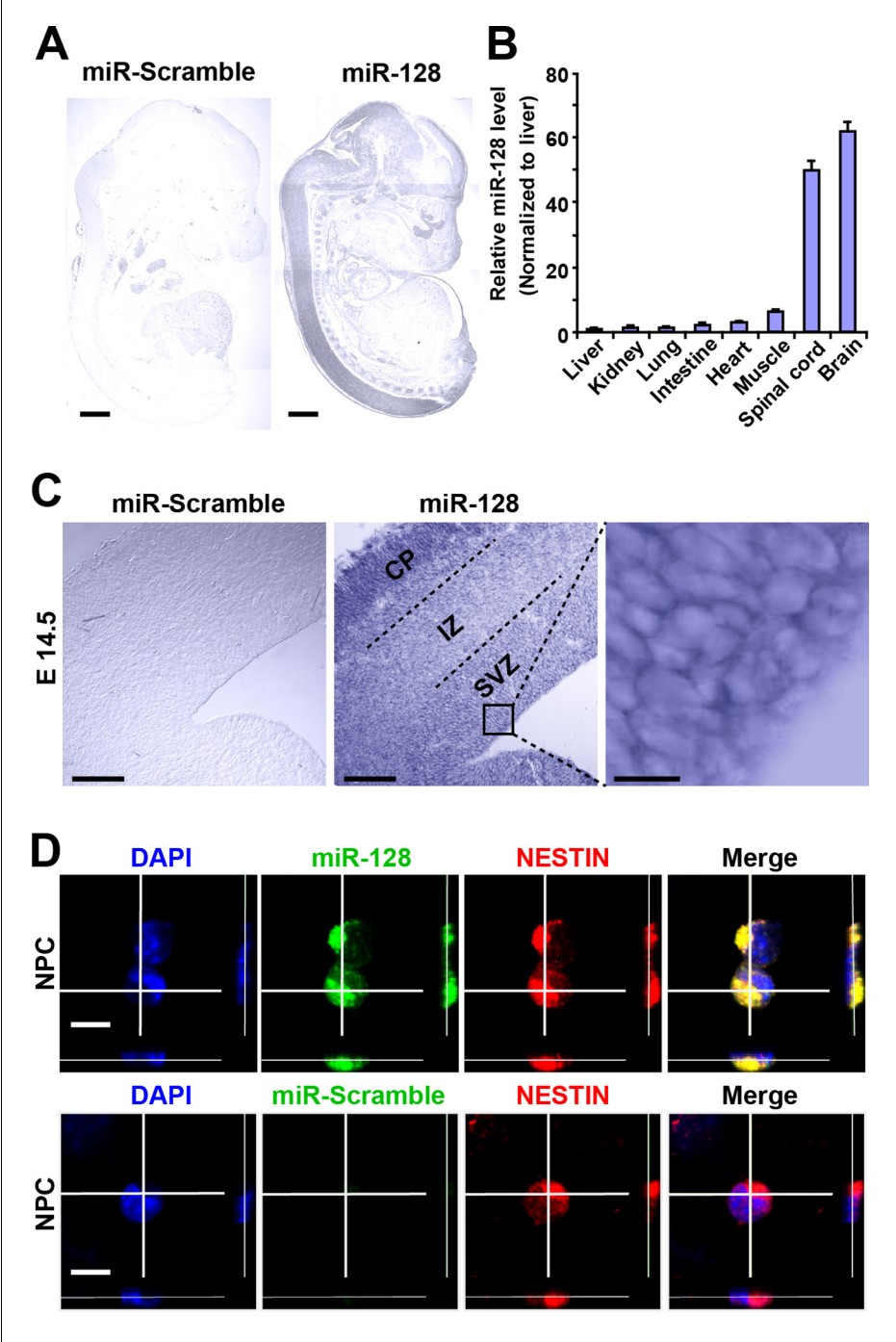

**Figure 1.** miR-128 expression in the developing cerebral cortex. (**A**) ISH was performed in an E17.5 mouse embryo with a miR-128 LNA detection probe. A sagittal section shows that miR-128 is expressed in the CNS. The left-side section was probed with a miR scramble control. Scale bars, 1 mm. (**B**) Real-time qPCR analyses of miR-128 from various tissues at E17.5, showing that miR-128 is highly expressed in the brain and spinal cord. miR-128 expression was measured and normalized to that of U6 and is shown relative to the liver expression level. The values represent the mean ± s.d. (n = 3). (**C**) miR-128 expression in the cortex. ISH was performed in an E14.5 mouse embryo brain coronal section with a miR-128 LNA detection probe. The left-side section was probed with a miR scramble control. Scale bars, 50 μm, and 5 μm in the higher magnification image (right-most panel). (**D**) miR-128 expression in NPCs in vitro. miR-128 LNA in situ hybridization followed by immunofluorescence analysis of the neural stem cell marker NESTIN in NPCs. DAPI staining indicates the location of cell nuclei. Scale bars, 10 μm.

The following figure supplement is available for figure 1:

**Figure supplement 1.** miR-128 expression in the developing CNS.

signal was detected with a scrambled miRNA probe (*Figure 1A*). As an alternative method, we performed quantitative real-time PCR (qPCR) and found spatial expression patterns of miR-128 that were similar to those observed using ISH (*Figure 1B*). Within the E14.5 forebrain, miR-128 was clearly detectable in the cortical layers, and high-magnification images of cortical slices at E14.5 revealed the expression of miR-128 in cells within the ventricular/subventricular zone (VZ/SVZ) (*Figure 1C*). To further confirm this, we performed fluorescence ISH in combination with immunostaining using the NPC marker NESTIN in cortical slices at E14.5 and found that NPCs within the VZ/SVZ expressed miR-128 (*Figure 1—figure supplement 1*). Furthermore, NPCs isolated from the E14.5 forebrain co-expressed miR-128 and NESTIN (*Figure 1D*), indicating the potential functional role of miR-128 in regulating the proliferation and/or differentiation of NPCs.

## miR-128 regulates the proliferation and differentiation of NPCs in vitro

To examine whether miR-128 regulates the proliferation and/or differentiation of embryonic NPCs, we designed constructs for gain-of-function and loss-of-function studies. A dual-promoter expression construct carrying CMV-driven mouse miR-128-1 precursor and EF1α-driven copGFP (*Hollis et al., 2009*) (miR-128) was used to overexpress miR-128, while a dual-promoter expression construct containing H1-driven anti-sense miR-128 short-hairpin RNA and CMV-driven copGFP (miR-Zip-128) (*Guibinga et al., 2012*) was used to knock down endogenous miR-128 (*Figure 2—figure supplement 1A and B*). When we transfected primary embryonic NPCs that were isolated from E14.5 mouse forebrains with these constructs, the miR-128-1 precursor was efficiently processed to generate the mature miRNA, as shown by a 15-fold upregulation of miR-128 in transfected cells (*Figure 2—figure supplement 1A*). In addition, miR-Zip-128 transfection resulted in efficient knockdown of endogenous miR-128 (*Figure 2—figure supplement 1B*).

NPCs that were isolated from E14.5 mouse forebrains were electroporated with the aforementioned constructs and were pulse-labeled with 5-bromo-2'-deoxyuridine (BrdU) for 6 hr to label dividing cells of a heterogeneous group of NPCs from all phases of the cell cycle (*Bez et al., 2003*; *Zhang et al., 2014*) (*Figure 2A*). Overexpression of miR-128 inhibited NPC proliferation, as indicated by a 56% reduction in BrdU incorporation in transfected cells (arrowheads) compared with the incorporation in those transfected with the miRNA mimic control (*Figure 2B and C*). Conversely, miR-128 knockdown led to a 50% increase in the number of GFP-BrdU double-positive cells compared to the scramble control (*Figure 2D and E*).

To determine whether modulating the levels of miR-128 affected cell death, we performed a TUNEL assay (*Cai et al., 2000*) (*Figure 2—figure supplement 2A–D*) and antibody staining against cleaved caspase-3 (*Chen and Dong, 2009*) (*Figure 2—figure supplement 2E–H*). These results showed that upregulation or downregulation of the miR-128 level specifically affected cell proliferation without affecting apoptosis.

Next, to determine the fate of the NPCs following cell-cycle exit, NPCs were transfected with the aforementioned constructs and were subsequently induced to differentiate in vitro by withdrawing growth factors from the culture medium for 5–6 days (*Ma et al., 2008*; *Zhang et al., 2014*) (*Figure 2F*). Neuronal differentiation was assayed by immunostaining with TUJ1, a specific antibody against beta-III-tubulin (*Ferreira and Caceres, 1992*). An increase in the number of GFP and TUJ1 double-positive cells upon treatment would indicate increased neuronal differentiation, whereas a decrease would indicate the inhibition of neurogenesis. Intriguingly, miR-128 overexpression increased the neuronal differentiation of NPCs, as shown by a significant increase (of approximately 100%) in the number of GFP and TUJ1 double-positive cells (*Figure 2G and H*). In contrast, miR-128 knockdown resulted in a 40% decrease of GFP-TUJ1 double-positive cells compared with treatment with the scrambled control miRNA (*Figure 2I and J*). When we assayed neuronal differentiation using immunostaining with the marker of mature neurons MAP2, we observed a similar effect on neuronal differentiation (*Figure 2—figure supplement 3A–D*). Moreover, lentiviral transduction of miR-128 or miR-Zip-128, as an alternative gene delivery approach, resulted in similar effects on neurogenesis (*Figure 2—figure supplement 4A–D*). Taken together, these results indicate that miR-128 overexpression enhances neuronal differentiation of NPCs following cell-cycle exit, while miR-128 knockdown shows the opposite effect on neuronal differentiation.

To further test whether the enhanced differentiation was restricted to a neuronal fate, we immunostained NPCs with a specific antibody against GFAP, a marker for glial cells, and observed that miR-128 overexpression significantly decreased the number of GFP and GFAP double-positive cells,

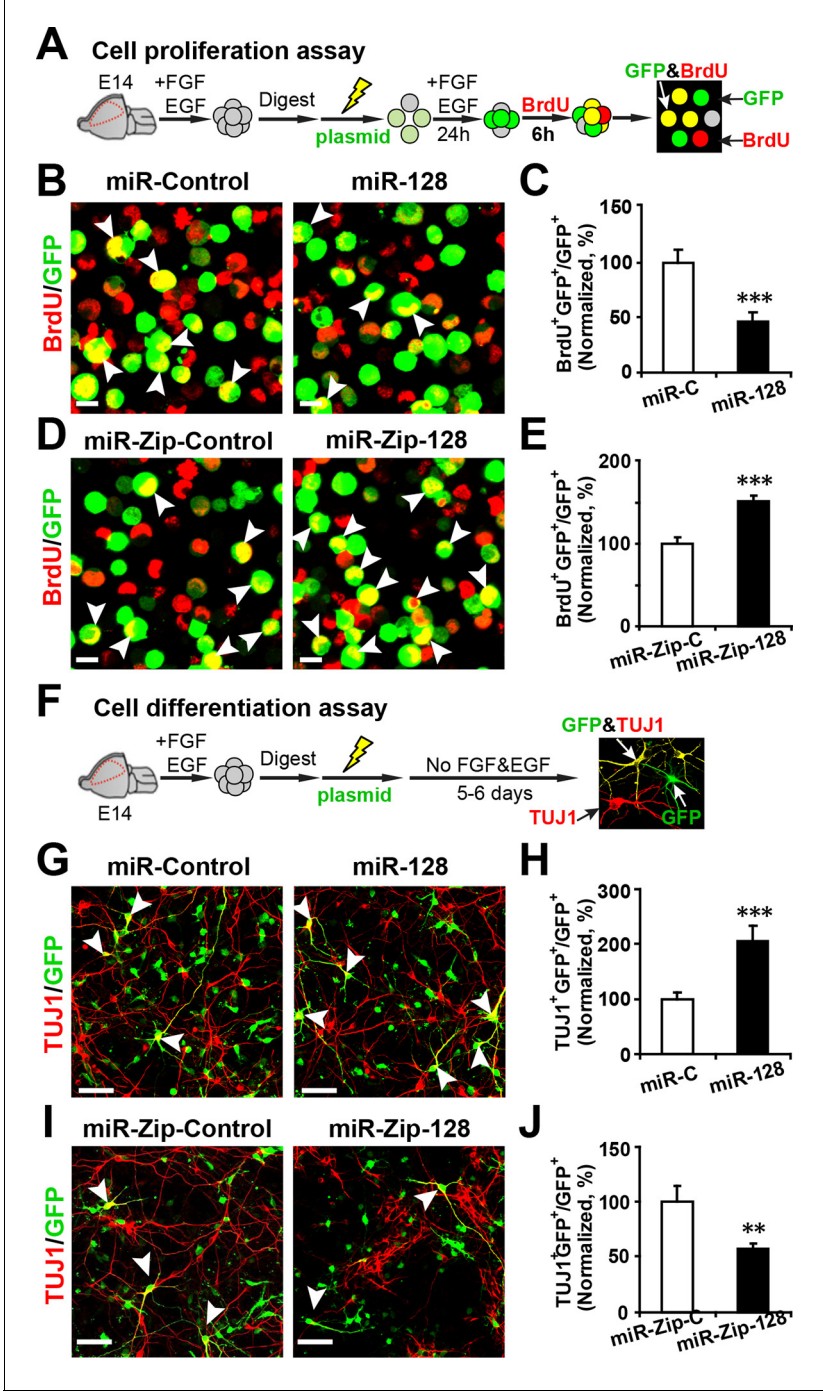

**Figure 2.** miR-128 modulates the proliferation and differentiation of NPCs in vitro. (**A**) Schematic representation of the cell proliferation assay procedures. (**B-E**) Ectopic expression of miR-128 decreases proliferation, while that of miR-Zip-128 increases NPC proliferation. NPCs were electroporated with the indicated plasmids and pulse-labeled with BrdU for 6 hr. NPCs were immunostained with an antibody against BrdU. The arrowheads indicate BrdU and GFP double-positive cells. Scale bars, 10 μm. Quantification of the number of GFP-BrdU double-positive cells relative to the total number of GFP-positive cells (**C, E**). (**F**) Schematic representation of the cell differentiation assay procedures. (**G-J**) Ectopic expression of miR-128 increases neurogenesis, while that of miR-Zip-128 decreases the neurogenesis of NPCs. NPCs were electroporated with the indicated plasmids and immunostained with antibodies against TUJ1. The arrowheads indicate TUJ1[+]GFP[+] cells. Scale bars, 50 μm. Quantification of the number of the GFP-TUJ1 double-positive cells relative to the number of GFP-positive cells (**H, J**). More than 1500 GFP-positive cells were counted for each condition. At least three sets of independent experiments were performed. The values represent the mean ± s.d. (n = 3). Student's *t*-test, differences were considered significant at ***p<0.001 and **p<0.01.

*Figure 2 continued on next page*

*Figure 2 continued*

The following figure supplements are available for figure 2:

**Figure supplement 1.** miR-128 overexpression and miR-128 knockdown constructs and their expression efficiency.

**Figure supplement 2.** miR-128 overexpression and knockdown does not affect NPC apoptosis.

**Figure supplement 3.** Ectopic expression of miR-128 increases neurogenesis, while that of miR-Zip-128 decreases neurogenesis.

**Figure supplement 4.** Lentivirus-mediated transduction of miR-128 and miR-Zip-128 increases and decreases neurogenesis, respectively.

**Figure supplement 5.** Ectopic expression of miR-128 decreases gliogenesis, while that of miR-Zip-128 increases gliogenesis.

**Figure supplement 6.** Optimizing the duration of BrdU pulse labeling for in vitro NPC proliferation assay.

indicating that NPCs that had exited the cell cycle following miR-128 overexpression may have over-committed to neuronal differentiation, preventing NPCs from becoming glial cells. The opposite effect was observed upon miR-128 knockdown in NPCs (*Figure 2—figure supplement 5A–D*).

To rule out the potential bias in the in vitro cell proliferation assay we monitored the time-course of BrdU incorporation using control NPCs (*Figure 2—figure supplement 6A*) as well NPCs electroporated with miR-ZIP-128. We found that the effects of miR-128 knockdown in NPCs (increased cell proliferation compared to control) remained unchanged at most of the time points (*Figure 2—figure supplement 6B*).

## miR-128 promotes neuronal differentiation in the developing cortex in vivo

During development, NPCs in the VZ maintain proliferative capacity and the ability to self-renew (apical progenitors, APs). These APs give rise to intermediate progenitors (basal progenitors, BPs) within the SVZ and IZ that can proliferate and become neurons (*Martinez-Cerdeno et al., 2006*; *Pontious et al., 2008*). To examine the role of miR-128 during neocortical development in vivo, we introduced miR-128 and miR-Zip-128 into the lateral ventricular wall of E13.5 mouse brains by in utero electroporation and analyzed the electroporated brains at E14.5.

First, to detect changes in NPC proliferation, we monitored the mitotic spindle orientation (*Huttner and Kosodo, 2005*; *Wang et al., 2011*) of APs within the VZ that were undergoing mitosis using an antibody against phosphorylated histone H3 (PH3) (*Postiglione et al., 2011*), which labels dividing nuclei (*Figure 3A*). We found a significant decrease in the percentage of horizontal divisions upon miR-128 overexpression (*Figure 3A and B*, 10% ± SD), while miR-128 knockdown led to a significant increase in this percentage (*Figure 3A and C*, 20% ± SD). Based on these observations, we performed further experiments to identify the fate of AP progeny following miR-128 overexpression and miR-128 knockdown.

Furthermore, miR-128 overexpression led to a marked decrease in the percentage of cells that were positively labeled for the incorporation of 5-ethynyl-2′-deoxyuridine (EdU) (*Ishino et al., 2014*) (~12%) (*Figure 3D and E*), reduced cell division, as indicated by immunostaining with Ki67 (*Figure 3—figure supplement 1A and B*), and a marked decrease in the number of cells that were positive for the AP marker PAX6 (28% reduction in the number of VZ/SVZ cells that were positive for both GFP and PAX6) (*Figure 3H and I*) and SOX2 (*Figure 3—figure supplement 2A and B*). These data indicate that miR-128 overexpression decreased the number of proliferating APs within the VZ/SVZ. In contrast, miR-128 knockdown had the opposite effects on EdU incorporation and on Ki67, PAX6 and SOX2 immunostaining in APs (*Figure 3F,G,J and K*) (*Figure 3—figure supplement 1C and D*, *2C and D*).

Next, given that we observed increased numbers of obliquely dividing cells upon miR-128 overexpression (*Figure 3A and B*), indicating potential expansion of BPs (*Huttner and Kosodo, 2005*; *Wang et al., 2011*), we monitored TBR2 expression upon miR-128 overexpression. MiR-128 overexpression led to an increase in the number of TBR2-positive cells (60% increase in GFP-TBR2 double-positive cells) (*Figure 3L and M*), while miR-128 knockdown resulted in a decrease in the number of

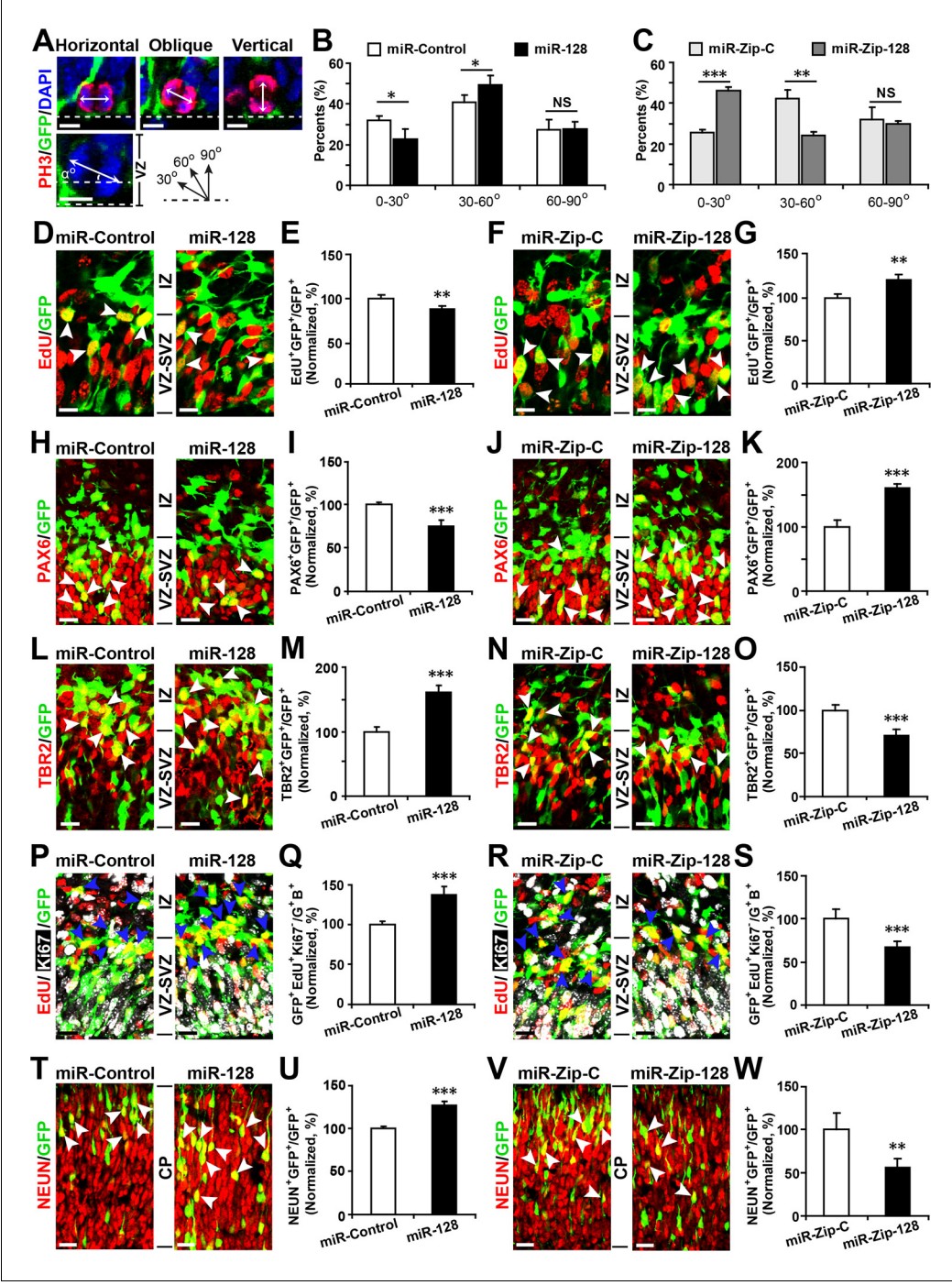

**Figure 3.** miR-128 regulates the proliferation and differentiation of NPCs in vivo. (A-C) miR-128 regulates the symmetric division of apical progenitors in the VZ/SVZ. Mouse embryos were electroporated at E13.5 with the indicated plasmids and sacrificed at E14.5. The nuclei of mitotic cells were labeled using antibodies against PH3. The orientation of the mitotic spindle relative to the ventricular surface was determined and categorized into horizontal (0–30°), oblique (30–60°), or vertical (60–90°) as pictured. Scale bars, 5 μm. (A). Percentage of GFP and PH3 double-positive cells with the indicated mitotic spindle orientation following miR-128 overexpression (B) and miR-128 knockdown (C). (D-G) miR-128 regulates the proliferation of apical progenitors in the VZ/SVZ. Mouse embryos were electroporated at E13.5 with the indicated constructs. Twenty-four hours post-electroporation, dividing cells were marked by EdU pulse-labeling for 2 hr and sacrificed. The arrowheads indicate EdU⁺GFP⁺ cells. Scale bars, 10 μm. Quantification of the number of GFP-EdU double-positive cells relative to the number of GFP-positive cells (E, G). (H–K) Ectopic expression of miR-128 decreases apical progenitors, while that of miR-Zip-128 increases apical progenitors. Mouse embryos were electroporated at E13.5 with the indicated constructs and sacrificed at E14.5. Brain sections were immunostained with antibodies against PAX6. The arrowheads indicate PAX6⁺GFP⁺ cells. Scale bars, 10 μm. Quantification of the number of GFP-PAX6 double-positive cells relative to the number of GFP-positive cells (I, K). (L–O) Ectopic expression of miR-128

*Figure 3 continued on next page*

*Figure 3 continued*

increases basal progenitors, while that of miR-Zip-128 decreases basal progenitors. Mouse embryos were electroporated at E13.5 with the indicated constructs and sacrificed at E14.5. Brain sections were immunostained with antibodies against TBR2. The arrowheads indicate TBR2$^+$GFP$^+$ cells. Scale bars, 10 μm. Quantification of the number of GFP-TBR2 double-positive cells relative to the number of GFP-positive cells (M, O). (P-S) miR-128 promotes cell cycle exit, and miR-Zip-128 inhibits cell cycle exit. Mouse embryos were electroporated at E13.5 with the indicated constructs. Twenty-four hours post-electroporation, dividing cells were marked by EdU pulse-labeling for 24 hr and sacrificed. The brain sections were immunostained with antibodies against Ki67. The blue arrowheads indicate GFP$^+$EdU$^+$Ki67$^-$ cells that had exited the cell cycle. Scale bars, 10 μm. Quantification of the number of GFP-EdU double-positive, Ki67-negative cells relative to the number of GFP-BrdU double-positive cells (Q, S). (T-W) miR-128 promotes the differentiation of NPCs in vivo. Mouse embryos were electroporated at E13.5 with the indicated constructs and sacrificed at E17.5. Brain slices were immunostained for NEUN. Arrowheads indicate NEUN$^+$GFP$^+$ cells. Scale bars, 10 μm. Quantification of the number of GFP-NEUN double-positive cells relative to the number of GFP-positive cells (U, W). More than 1500 GFP-positive cells were counted for each condition. At least three sets of independent experiments were performed. The values represent the mean ± s.d. (n = 3). Student's t-test, differences were considered significant at ***p<0.001, **p<0.01, and *p<0.05.

The following figure supplements are available for figure 3:

**Figure supplement 1.** Ectopic expression of miR-128 decreases proliferation, while that of miR-Zip-128 increases proliferation in vivo.

**Figure supplement 2.** Ectopic expression of miR-128 decreases apical progenitors, while that of miR-Zip-128 increases apical progenitors.

**Figure supplement 3.** Model of the miR-128 in vivo phenotype.

TBR2-positive cells (30% decrease in GFP-TBR2 double-positive cells) (*Figure 3N and O*). Taken together, these data indicate that miR-128 regulates NPC proliferation by promoting intermediate basal progenitors at the expense of apical progenitors within the VZ/SVZ.

Because BPs will generate the bulk of cortical neurons (*Tan and Shi, 2013*), we tested whether miR-128 regulates the neuronal differentiation of NPCs in vivo. First, we analyzed the number of BPs that exited the cell cycle (*Ge et al., 2010*; *Yang et al., 2012*) upon miR-128 manipulation. Following in utero electroporation, dividing cells were labeled with EdU for 24 hr and subsequently immunostained for Ki67. Compared to the control treatment, miR-128 overexpression increased the number of cells that were positive for both GFP and EdU but negative for Ki67 cells (by ~40%), indicating an increase the number of BPs that had exited the cell cycle (*Figure 3P and Q*). Conversely, miR-128 knockdown had the opposite effect (*Figure 3R and S*). To further examine whether the observed changes in cell cycle exit ultimately led to a difference in neurogenesis, we analyzed brains at E17.5, four days after electroporation, and assayed neuronal differentiation by immunostaining with a specific antibody against the neuronal marker NEUN (*Zhang et al., 2014*). miR-128 overexpression significantly increased (by ~25%) the number of NEUN-positive neurons in the CP zone compared with the number in the control condition (*Figure 3T and U*). Moreover, miR-128 knockdown had the opposite effect on neurogenesis (*Figure 3V and W*). Taken together, these results indicate that miR-128 may act on two different stages of NPC development: first, by regulating symmetric/asymmetric division of APs, miR-128 promotes BP production; and second, by enhancing the exit of BPs from the cell cycle, miR-128 promotes overall neurogenesis (*Figure 3—figure supplement 3*). In contrast, downregulation of miR-128 in early neuronal precursors impeded their developmental progression by causing them to be retained in a more primitive, proliferative stage, resulting in the expansion of a pool of the NPC pool. Intriguingly, this early expansion of NPC pools upon miR-128 knockdown did not result in a net increase in neuronal numbers at E17.5, suggesting that further investigation is necessary to delineate whether the observed phenomena is due to compromised neurogenic capability of NPCs or simply due to a delay in neurogenesis which could eventually be overcome at a later postnatal stage (*Figure 3—figure supplement 3*).

## PCM1 is a direct target of miR-128 in NPCs

miRNAs normally regulate the translation and/or degradation of multiple target mRNAs (*Bartel, 2009*). To further characterize the molecular mechanisms that underlie the changes in NPC competence, we utilized two widely used in silico microRNA target prediction algorithms (TargetScan and miRanda) (*Mi et al., 2013*; *Witkos et al., 2011*), which identified 800 and 940 potential targets of miR-128, respectively (*Figure 4—source data 1*). Among the 77 overlapping targets identified

using two algorithms, only 53 genes were annotated to have known biological processes and thus selected for further testing (*Figure 4—source data 1*). qPCR analysis of cultured NPCs that overexpressed miR-128 revealed 21 genes that were downregulated (*Figure 4—source data 2*). Eleven out of the 53 genes tested exhibited consistent reduction in mRNA levels upon miR-128 overexpression in cultured mouse NPCs (*Pcm1, Lmbr1l, Foxo4, Sh2d3c, Nfia, Pde8b, Sec24a, Pde3a, Fbxl20, Ypel3* and *Kcnk10*, *Figure 4—source data 2*, highlighted in yellow). The 11 genes were further tested for reciprocal upregulation when miR-128 was inhibited. qPCR analysis following miR-128 inhibition showed that only *Pcm1, Nfia, Foxo4*, and *Fbxl20* were consistently upregulated among which Pcm1 displayed the greatest change (*Figure 4—source data 3*).

We further validated *Pcm1* as a target of miR-128 using a luciferase assay. First, we cloned the 3'-UTR of *Pcm1* (WT-*Pcm1*) into a dual-luciferase reporter construct, pmirGLO, to assess translation of the target protein based on the luciferase activity (*Krishnan et al., 2013*) (*Figure 4A*). In this assay, co-transfection of miR-128 with the WT-*Pcm1* reporter construct markedly suppressed the luciferase activity (by 58%, *Figure 4B*). However, co-transfection of miR-128 with random 3'-UTR sequences (Control, *Figure 4B*) did not affect the luciferase activity. To further determine whether the targeting of PCM1 by miR-128 was specific, we introduced three mismatched nucleotides to the predicted seed region of the miR-128 binding site (MT-*Pcm1*) (*Figure 4A*, red underlines). Mutating these seed sequences abolished the miR-128-mediated suppression of PCM1 luciferase activity and restored the luciferase activity to the control level (*Figure 4B*), indicating the specificity of miR-128 targeting of the 3'-UTR of *Pcm1*.

Next, we examined whether miR-128 downregulated the endogenous expression of PCM1 at the mRNA and protein levels by overexpressing either miR-128 or a scrambled control in NPCs and then performing qPCR and western blot analyses using a specific antibody against PCM1. We observed that miR-128 overexpression significantly reduced PCM1 mRNA (*Figure 4C*) and protein levels (*Figure 4D*). Conversely, miR-128 knockdown using a specific inhibitor of miR-128 (anti-miR-128) (*Smrt et al., 2010*), in comparison to using a scrambled anti-miR (anti-miR-control) (*Figure 4—figure supplement 2*) in NPCs produced the opposite effect on the expression of PCM1 mRNA (*Figure 4E*) and protein (*Figure 4F*). Taken together, these data suggest that miR-128 targets PCM1 expression in NPCs, which in turn controls NPC proliferation and differentiation in vitro. In addition, qPCR analyses of tissue samples that were isolated from the VZ/SVZ, IZ, and CP using laser capture microdissection (LCM) (*Wang et al., 2009*) revealed an inverse relationship between miR-128 and PCM1 mRNA (*Figure 4G and H*). An inverse relationship between miR-128 and PCM1 mRNA levels was also evident temporally, given that the expression of miR-128 gradually increased starting from E12.5 through P0, whereas PCM1 protein expression was found to gradually decrease over this time period (*Figure 4—figure supplement 3A–B*), suggesting that miR-128 might regulate PCM1 levels to control NPC proliferation and differentiation in the developing cortex.

## PCM1 regulates the proliferation and differentiation of NPCs

If the effect of miR-128 on the proliferation and differentiation of NPCs is mediated through the suppression of endogenous PCM1, then PCM1 downregulation should mimic the cellular effects of miR-128 overexpression. To test this hypothesis, we generated two small hairpin RNA (shRNA) vectors that expressed shRNAs that targeted mouse PCM1 and validated the efficiency of these shRNAs in knocking down PCM1 in mouse neuroblastoma (N2A) cells (*Figure 5—figure supplement 1A*). The expression of the shRNA vectors #1 and #2 led to a reduction in endogenous PCM1 of approximately 60 and 70%, respectively. Based on these data, we used shRNA #2 to examine the role of PCM1 in early neurogenesis.

PCM1 has been shown to affect the proliferation and neurogenesis of NPCs; knockdown of PCM1 inhibits NPC proliferation but promotes NPC differentiation in the developing cortex (*Ge et al., 2010*). To further confirm the loss of PCM1 function in NPCs, we electroporated NPCs with PCM1 shRNA and assessed the proliferation of NPCs using BrdU pulse-labeling for 6 hr. The reduction of endogenous PCM1 via shRNA led to significant inhibition of NPC proliferation, as indicated by a 35% reduction in BrdU incorporation by transfected cells compared with the BrdU incorporation by NPCs that were transfected with a control scrambled shRNA (*Figure 5A and 5B*); this reduction was comparable to that caused by the overexpression of miR-128 in NPCs (*Figure 2B and C*). PCM1 knockdown did not affect cell survival, as indicated by TUNEL assay and antibody staining against activated caspase-3 (*Figure 5—figure supplement 2A–D*).

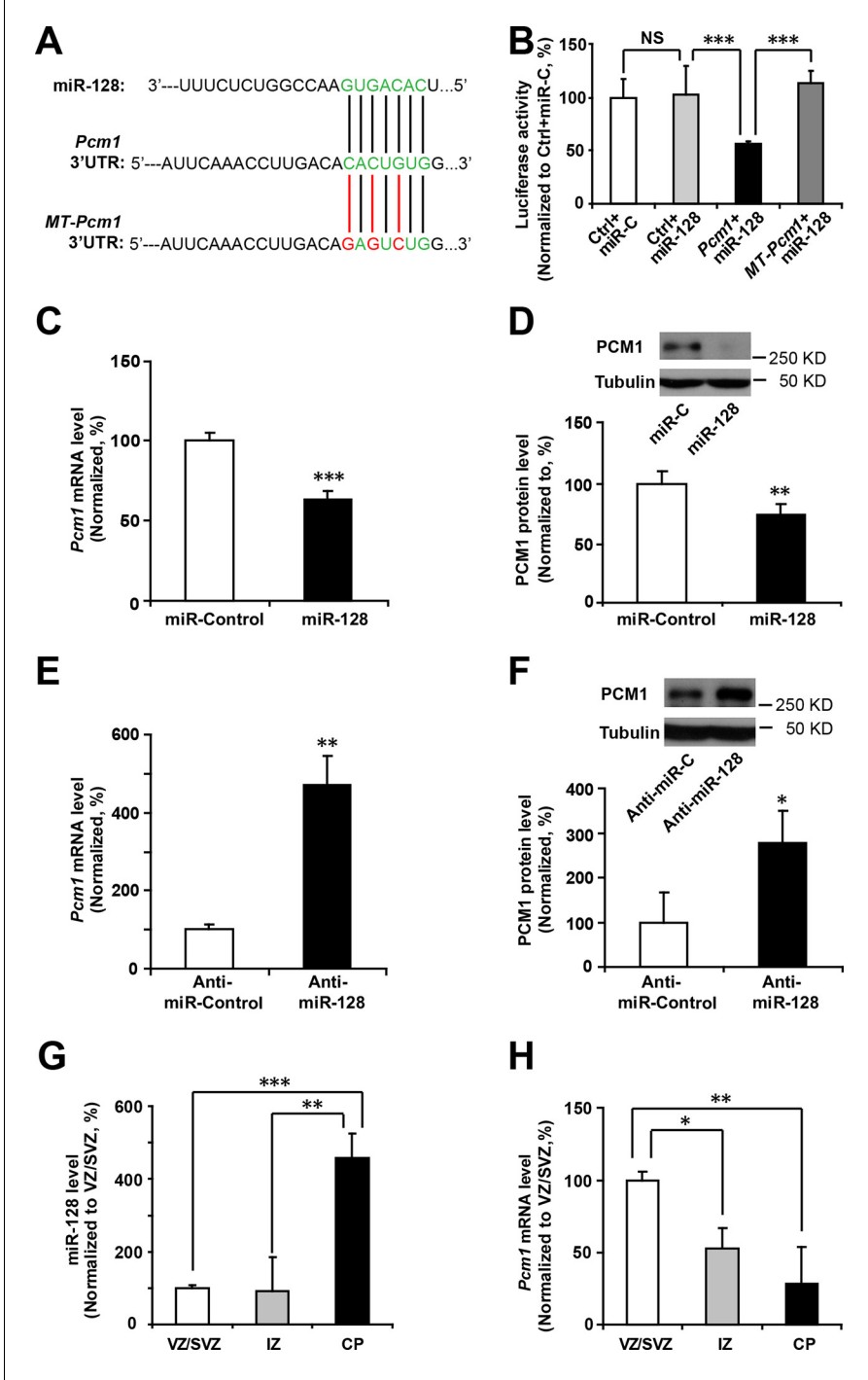

**Figure 4.** miR-128 regulates PCM1 expression in NPCs. (A) TargetScan analysis identified a miR-128 target site in the mouse *Pcm1* 3'-UTR region (highlighted in green). The mutant PCM1 is shown, with the seed binding sites highlighted in red. (B) PCM1 luciferase activity is suppressed by miR-128. HEK293T cells were co-transfected with miR-128 and the 3'-UTR of *Pcm1* containing either the miRNA binding site (WT) or mutant (MT) versions of the *Pcm1* seed binding sites for 2 days. The cells were harvested and lysed, and a luciferase activity assay was then performed. miR-128-mediated suppression of PCM1 luciferase activity was relieved upon mutation of the *Pcm1* seed binding sites. (C,D) miR-128 overexpression in NPCs led to reduced endogenous *Pcm1* mRNA levels, as determined by qPCR (C), and PCM1 protein expression, as demonstrated via densitometry analysis of western blots (D). (E,F) anti-miR-128 leads to increased endogenous *Pcm1* mRNA levels, as demonstrated by qPCR (E), and protein expression of PCM1 (F). (G,H) LCM was used to isolate RNA from three specific cortical layers of E14.5 embryonic brains: the VZ/SVZ, IZ, and CP. qPCR quantification of miR-128 levels (G) and *Pcm1* mRNA levels (H). At least three sets of independent experiments were performed. The values represent the mean ± s.d.
*Figure 4 continued on next page*

*Figure 4 continued*

(n = 3). Student's *t*-test, differences were considered significant at ***p<0.001, **p<0.01, and *p<0.05 for all panels in the figure. ANOVA, differences were considered significant at ***p<0.001 and **p<0.01.

The following source data and figure supplements are available for figure 4:

**Source data 1.** Gene Ontology (GO) of the miR-128 target gene list.

**Source data 2.** Relative expression of 53 predicted miR-128 targets upon overexpression of miR-128 in NPCs.

**Source data 3.** Relative expression of miR-128 targets upon downregulation of miR-128 in NPCs using miR-Zip-128.

**Source data 4.** List of qPCR primers.

**Figure supplement 1.** Schematic diagram outlines rationale of gene selection process.

**Figure supplement 2.** miR-128 inhibitor knockdown efficiency.

**Figure supplement 3.** Inverse relationship between the temporal expression patterns of miR-128 and PCM1.

Next, to determine whether knocking down endogenous PCM1 could trigger the neuronal differentiation of NPCs, NPCs were transfected with PCM1 shRNA or a scrambled control shRNA, induced to differentiate, and immunostained for TUJ1 (*Figure 5C and D*). Similar to miR-128 overexpression, PCM1 shRNA expression significantly increased the neuronal differentiation of NPCs (by approximately 60% compared with that of scrambled shRNA-transfected NPCs, *Figure 5C and D*). Consistent with this finding, MAP2 immunostaining revealed a similar increase in the neuronal differentiation of PCM1 shRNA-expressing NPCs (*Figure 5—figure supplement 3A and B*). Taken together, these results indicate that inhibiting PCM1, which is a target of miR-128, mimics the cellular effect of miR-128 on NPC proliferation and differentiation.

To examine whether PCM1 overexpression would exert the opposite effect of PCM1 knockdown in NPCs, we co-expressed PCM1 (without its 3'-UTR) in NPCs using a GFP expression construct and found that PCM1 overexpression increased the proliferation of NPCs compared to the vector-only control treatment (70% increase in BrdU and GFP double-positive cells) (*Figure 5—figure supplement 4A and B*). However, PCM1 overexpression decreased neurogenesis, as determined by immunostaining for TUJ1 (35% decrease in the number of TUJ1 and GFP double-positive cells) (*Figure 5—figure supplement 5A and B*) and MAP2 (45% decrease in the number of MAP2 and GFP double-positive cells) (*Figure 5—figure supplement 5C and D*). TUNEL and activated caspase-3 staining demonstrated that PCM1 overexpression had no effect on apoptosis (*Figure 5—figure supplement 6*). We confirmed these in vitro findings in vivo by electroporating PCM1 into NPCs in the VZ of E13.5 mouse brains. First, we monitored the mitotic spindle orientation of APs within the VZ that were undergoing mitosis by labeling the dividing nuclei with PH3 (*Figure 5—figure supplement 7A*). We observed a significant increase in the percentage of horizontally dividing cells upon PCM1 overexpression (*Figure 5—figure supplement 7B*, 20% ± SD). In addition, the expression of exogenous PCM1 in vivo led to a marked increase in the number of cells that incorporated EdU (40% increase in EdU and GFP double-positive cells) (*Figure 5—figure supplement 7C and D*). Furthermore, we observed a marked increase in the number of PAX6-positive apical progenitor cells (60% increase in PAX6 and GFP double-positive cells) (*Figure 5—figure supplement 7E and F*) and a decrease in the number of TBR2-positive intermediate progenitor cells in brains in which NPCs had been electroporated with PCM1 (40% decrease in TBR2 and GFP double-positive cells) (*Figure 5—figure supplement 7G and H*). We monitored the neuronal differentiation of PCM1-expressing NPCs in vivo using NEUN staining and observed a 25% decrease in the neuronal differentiation of the NPCs (*Figure 5—figure supplement 7I and J*).

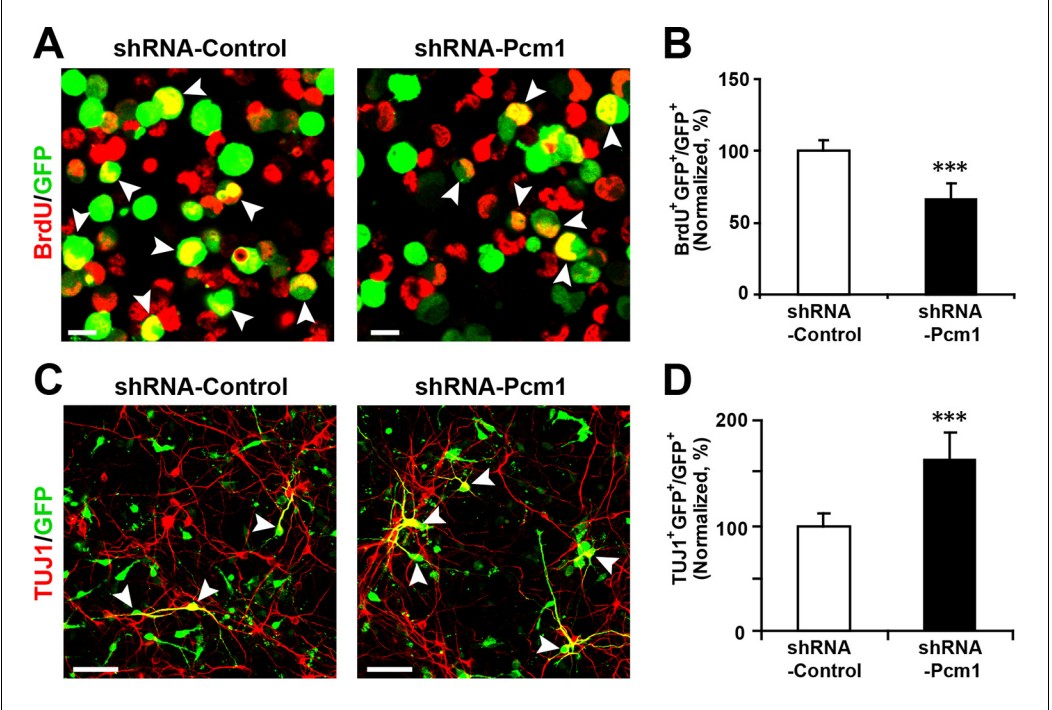

**Figure 5.** PCM1 knockdown decreases NPC neurogenesis. (A,B) PCM1 knockdown in NPCs decreases neural proliferation. NPCs were electroporated with a plasmid expressing PCM1 shRNA or with a control vector and pulse-labeled with BrdU for 6 hr. NPCs were immunostained with an antibody against BrdU. The arrowheads indicate BrdU and GFP double-positive cells. Scale bars, 10 μm. Quantification of the number of GFP-BrdU double-positive cells relative to the total number of GFP-positive cells (B). (C,D) PCM1 knockdown in NPCs increases neurogenesis. NPCs were electroporated with a plasmid expressing PCM1 shRNA or with a control vector and then immunostained with antibodies against TUJ1. The arrowheads indicate TUJ1$^+$GFP$^+$ cells. Scale bars, 50 μm. Quantification of the number of GFP-TUJ1 double-positive cells relative to the number of GFP-positive cells (D). More than 1500 GFP-positive cells were counted for each condition. At least three sets of independent experiments were performed. The values represent the mean ± s.d. (n = 3). Student's $t$-test, differences were considered significant at ***p<0.001.

The following figure supplements are available for figure 5:

**Figure supplement 1.** Efficient knockdown of PCM1.

**Figure supplement 2.** PCM1 knockdown in NPCs did not trigger apoptotic cell death.

**Figure supplement 3.** PCM1 knockdown in NPCs increases neurogenesis.

**Figure supplement 4.** Overexpression of PCM1 increases NPC proliferation.

**Figure supplement 5.** Overexpression of PCM1 decreases NPC neuronal differentiation.

**Figure supplement 6.** Overexpression of PCM1 in NPCs did not trigger apoptotic cell death.

**Figure supplement 7.** PCM1 regulates NPC proliferation and differentiation in vivo.

## PCM1 overexpression rescues the observed effects of miR-128 overexpression on NPC proliferation and differentiation

To determine whether the effect of miR-128 on neuronal differentiation is mediated through PCM1, we co-electroporated cells with miR-128 and a PCM1 expression vector lacking the 3'-UTR. A vector-only construct was used as the corresponding control construct for PCM1 overexpression. We found that PCM1 overexpression rescued the decrease in NPC proliferation (103% versus 62% for BrdU and GFP double-positive cells, *Figure 6A and B*), and reversed the observed increase in neuronal differentiation induced by miR-128 overexpression (150% versus 225% for TUJ1 and GFP

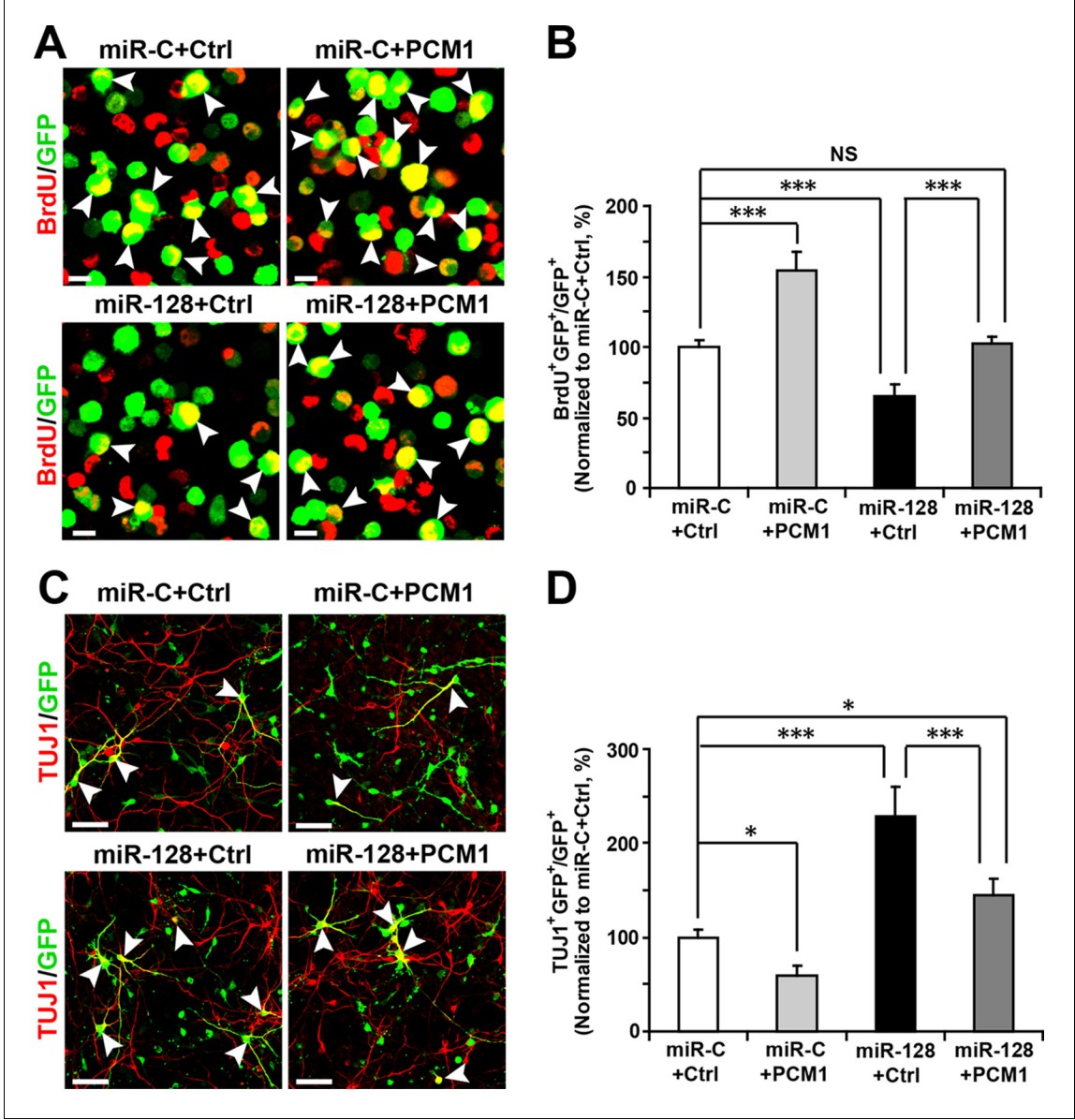

**Figure 6.** PCM1 is a functional target of miR-128 in vitro. (A,B) PCM1 antagonizes the effects of miR-128 on NPC proliferation in vitro. NPCs were electroporated with a miRNA control vector or with miR-128. Either a PCM1 expression construct or vector only control was co-electroporated in miR-128-expressing NPCs to examine the rescue effect of PCM1 in these cells. The cells were pulse-labeled with BrdU for 6 hr, and a proliferation assay was performed by immunostaining for BrdU (A). Scale bars, 10 μm. (B) Quantification of BrdU and GFP double-positive cells, demonstrating that PCM1 overexpression in the miR-128 cells rescued the reduced proliferation in the miR-128-expressing cells. (C,D) miR-128 antagonizes the function of PCM1 in neurogenesis in vitro. A neuronal differentiation assay was performed by immunostaining for TUJ1. The arrowheads indicate cells that are double-positive for TUJ1 and GFP (C). Scale bars, 50 μm. (D) Quantification of TUJ1 and GFP double-positive cells, demonstrating that PCM1 overexpression in the miR-128-expressing cells reverses the increased neurogenesis phenotype of the miR-128-expressing cells. More than 2000 GFP-positive cells were counted for each condition. At least three sets of independent experiments were performed. The values represent the mean ± s.d. (n = 3). ANOVA, differences were considered significant at ***p<0.001, **p<0.01 and *p<0.05.

The following figure supplement is available for figure 6:

**Figure supplement 1.** PCM1 is a functional target of miR-128 in vitro.

double-positive cells, *Figure 6C and D*; 115% versus 160% for MAP2 and GFP double-positive cells; *Figure 6—figure supplement 1A and B*) in vitro.

To further verify these results in vivo, we co-electroporated the PCM1 expression vector with miR-128 in utero. Similarly, we found that co-expression of PCM1 with miR-128 successfully reversed the effects of miR-128 on neural stem cell proliferation (100% versus 75% for EdU and GFP double-positive cells, *Figure 7A and B*), PAX6 (78% versus 60% for PAX6 and GFP double-positive cells, *Figure 7C and D*), and TBR2 expression (120% versus 160% for TBR2 and GFP double-positive cells, *Figure 7E and F*) as well as on neuronal differentiation (100% versus 123% for NEUN-positive cells, *Figure 7G and H*). Taken together, these results strongly suggest that miR-128 regulates the proliferation and differentiation of NPCs in the murine embryonic cortex by targeting PCM1 expression though a direct interaction with its 3'UTR.

## Discussion

The cerebral cortex, which is the most complex structure of the brain, is responsible for cognitive, motor and perceptual behaviors (*Volvert et al., 2012*). The generation of cortical neurons depends on NPCs exiting the cell cycle, migrating, and subsequently partially maturing into neurons. These processes are orchestrated by multiple gene products that ultimately converge on the cytoskeleton to support morphological remodeling (*Sun and Hevner, 2014*). Therefore, for all of these processes to be correctly executed, the timing and abundance of specific gene products must be precisely regulated, and a lack of regulation may result in the cortical malformations that are associated with certain neuropsychiatric disorders (*Zhang et al., 2014*).

miRNAs are abundant, short-lived double-stranded RNAs of ~20-25 nucleotides that are derived from endogenous short-hairpin transcripts (*Bartel, 2009*). miRNAs contribute to various developmental processes by acting as post-transcriptional regulators; thus, they introduce an additional level of intricacy to gene regulation in neurogenesis. Recent data obtained by several groups support a primary role of miRNAs in fine-tuning signaling pathways that control the synchronized steps of cortical development (*Kawahara et al., 2012*; *Shi et al., 2010*).

Abu-Elneel et al. investigated the expression of 466 human miRNAs from postmortem cerebellar cortical tissue from individuals with ASDs and identified twenty-eight miRNAs that were expressed at significantly different levels in the ASD brains compared with the non-autism control brains (*Abu-Elneel et al., 2008*). Interestingly, of these dysegulated miRNAs, only three of them (miR-7, miR-128, and miR-132) were exclusively expressed in the brain (*Li and Jin, 2010*).

miR-128 is transcribed from two distinct loci, miR-128-1 and miR-128-2, as two primary transcripts that are processed into identical mature miRNA sequences (*Adlakha and Saini, 2014*). miR-128-1 and miR-128-2 reside in the intronic regions of genes on two different chromosomes (*Tan et al., 2013*). Previously, downregulation of miR-128 has been reported in several brain cancers, including glioblastoma and medulloblastoma (*Adlakha and Saini, 2013*; *Peruzzi et al., 2013*). Consistent with these findings, allelic loss of chromosome 3p, where miRNA-128-2 is encoded, has also been associated with the most aggressive forms of neuroblastoma, indicating that miR-128 may play an important role in the cell cycle as well as growth and differentiation (*Adlakha and Saini, 2014*).

In identifying the targets of miR-128, we eventually narrowed our search of potential miR-128 targets in NPCs to *Pcm1*, *Nfia*, *Foxo4*, and *Fbxl20* (*Figure 4—source data 1*). Among them, *Foxo4*, which encodes for an insulin/IGF-1 responsive transcription factor that regulates cell cycles (*Furukawa-Hibi et al., 2005*; *Schmidt et al., 2002*), was ruled out as a probable functional target of miR-128 based on a recent study that reported the loss of FOXO4 reduces the potential of human embryonic stem cells (hESCs) to differentiate into neural lineages (*Vilchez et al., 2013*), which is opposite from miR-128 overexpression effects that we observed. *Nfia* (Nuclear Factor I/A) encodes for a protein that functions as a transcription and replication factor for adenovirus DNA replication (*Qian et al., 1995*), while *Fbxl20*, encodes for a F-box protein which is involved in synaptic plasticity of neuronal networks (*Takagi et al., 2012*). As such, both genes were ruled out since their known biological functions were not relevant to the neurogenesis phenotype observed in miR-128 manipulation. Based on these rationales, we sought to focus on PCM1 as our primary gene of interest and investigate the functional relationship between PCM1 and miR-128 in early neurogenesis. Furthermore, based on our expression analysis, miR-128 expression is specific to neural stem cells and gradually increases during cortical development (*Figure 1*). In contrast, the expression level of its target

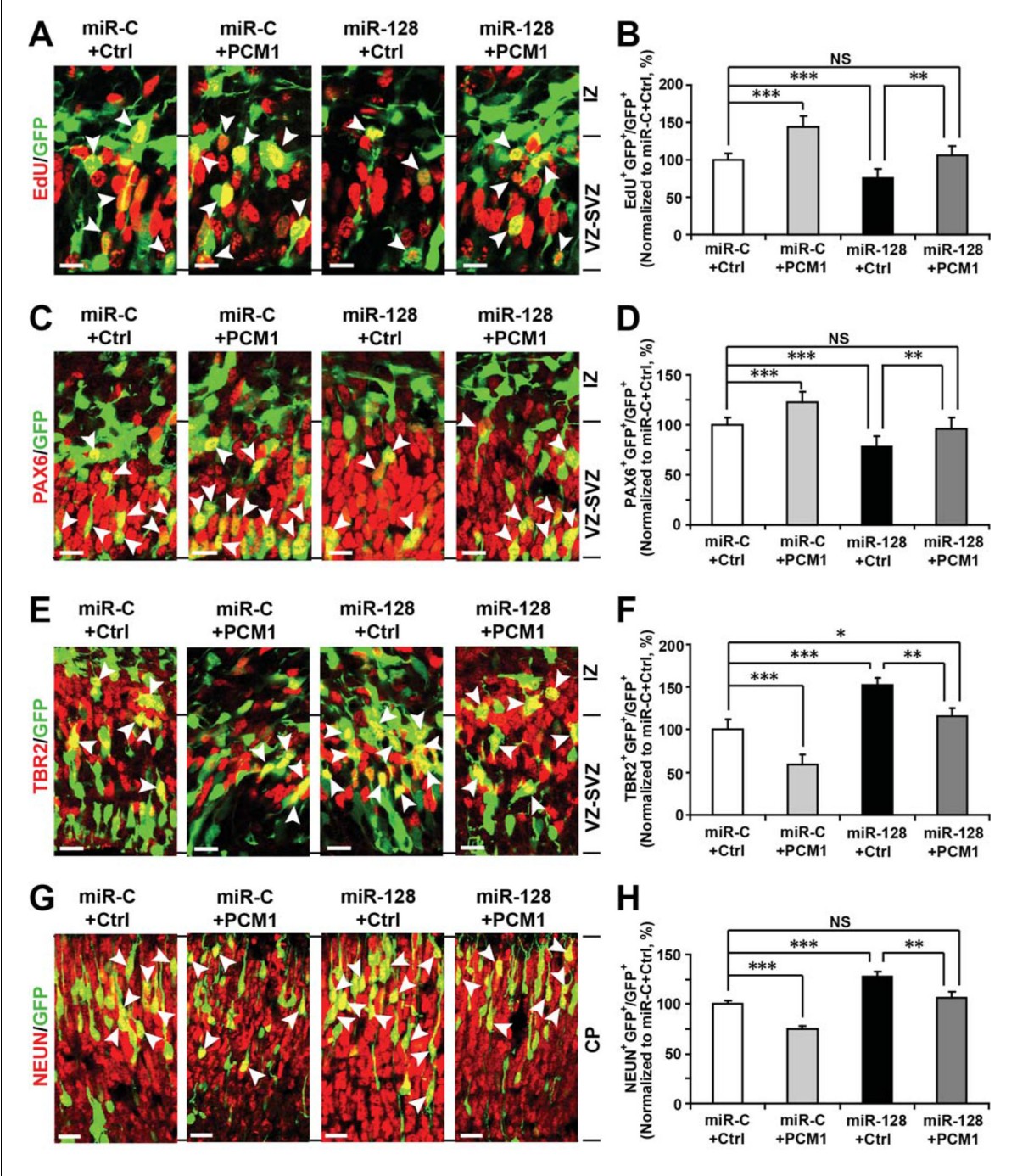

**Figure 7.** PCM1 is a downstream target of miR-128 during NPC proliferation and differentiation in vivo. (A,B) PCM1 antagonizes the effects of miR-128 on NPC proliferation in vivo. Mouse embryos were electroporated at E13.5 with a miRNA control vector or with miR-128. Either a PCM1 expression construct or vector only control was co-electroporated in miR-128-expressing NPCs to examine the rescue effect of PCM1 in these cells. Twenty-four hours post-electroporation, dividing cells were marked by EdU pulse-labeling for 2 hr and sacrificed. The arrowheads indicate EdU$^+$GFP$^+$ cells. Scale bars, 10 µm. Quantification of the number of GFP-EdU double-positive cells relative to the number of GFP-positive cells (B). (C,D) PCM1 overexpression rescues the miR-128-mediated decrease in the number of apical progenitors. Mouse embryos were electroporated at E13.5 with the indicated constructs and sacrificed at E14.5. Brain sections were immunostained with antibodies against PAX6. The arrowheads indicate PAX6$^+$GFP$^+$ cells. Scale bars, 10 µm. Quantification of the number of GFP-PAX6 double-positive cells relative to the number of GFP-positive cells (D). (E,F) PCM1 overexpression rescues the miR-128-mediated increase in the number of basal progenitors. Mouse embryos were electroporated at E13.5 with the indicated constructs and sacrificed at E14.5. Brain sections were immunostained with antibodies against TBR2. The arrowheads indicate TBR2$^+$GFP$^+$ cells. Scale bars, 10 µm. Quantification of the number of GFP-TBR2 double-positive cells relative to the number of GFP-positive cells (F). (G,H) PCM1

*Figure 7 continued on next page*

*Figure 7 continued*

overexpression rescues the miR-128-mediated increase in neurogenesis. Mouse embryos were electroporated at E13.5 with the indicated constructs and sacrificed at E17.5. Brain slices were immunostained for NEUN. Arrowheads indicate NEUN⁺GFP⁺ cells. Scale bars, 10 μm. Quantification of the number of GFP-NEUN double-positive cells relative to the number of GFP-positive cells (H). More than 1500 GFP-positive cells were counted for each condition. At least three sets of independent experiments were performed. The values represent the mean ± s.d. (n = 3). ANOVA, differences were considered significant at ***p<0.001 and **p<0.01.

protein PCM1 is higher during early developmental stages and lower during late developmental stages (*Figure 4—figure supplement 3*). The inverse expression patterns of PCM1 and miR-128 indicate that miR-128 may function by turning off PCM1 expression, indicating that PCM1 may be the primary regulator by which miR-128 governs NPC proliferation and differentiation.

In this study, we sought to recapitulate the miR-128 upregulation observed in some ASD brains by overexpressing miR-128 specifically in NPCs. Importantly, we observed a dramatic change in the proliferation and differentiation of NPCs. The observed effects of miR-128 are consistent with a previous study by Ge et al. that demonstrated that the loss of PCM1 triggered NPCs to exit the cell cycle early and promoted the premature differentiation of NPCs to neurons (*Ge et al., 2010*). In this study, knockdown of PCM1 resulted in impaired interkinetic nuclear migration of NPCs, which leads to the overproduction of neurons and to premature depletion of the NPC pool in the developing neocortex. Consistent with this description, our results showed that miR-128 overexpression produced a phenotype similar to that previously reported for PCM1 knockdown (*Figures 2* and *3*). More importantly, we found that this phenotype could be rescued by the co-expression of a miR-128-resistant PCM1 variant (*Figures 6* and *7*). In contrast, knockdown of miR-128 resulted in increased levels of PCM1 in NPCs (*Figures 4*) and showed the opposite phenotype of miR-128 overexpression in terms of NPC proliferation and differentiation (*Figures 2* and *3*). Our results suggest that miR-128 regulates NPC proliferation and differentiation by fine-tuning endogenous PCM1 levels, which serve as a primary regulator that is required for proper neurogenesis to occur.

Intriguingly, we recently identified 9 novel (i.e., previously unreported) missense mutations in the *Pcm1* gene in ASD patients (H.S.J. and S.G.R., unpublished observations), indicating that PCM1 misregulation might be a core mechanism in some ASD patients with disrupted cortical development. Other recent studies using miR-128-2 knockout mice indicate that miR-128 levels regulate the excitability of adult neurons (*Tan et al., 2013*). By selectively inactivating miR-128-2 in forebrain neurons using Camk2a-Cre and floxed miR-128-2, Tan et al. found that reduced miR-128 expression triggered the early onset of hyperactivity, seizures, and death (*Tan et al., 2013*). Based on their bioinformatics network and pathway analyses of miR-128 target genes, those authors found that miR-128 may regulate the expression of numerous ion channels and transporters as well as genes that contribute to neurotransmitter-driven neuronal excitability and motor activity (*Tan et al., 2013*). Because NPCs are not excitable due to a lack of active sodium channels (*Li et al., 2008*), it is unlikely that the cellular effects of miR-128 observed here resulted from changes in the expression of ion channels or transporters. However, it will be interesting to follow neurons derived from NPCs with misregulated miR-128 to characterize how these neurons integrate into and function in cortical circuits. Moreover, it will be interesting to generate miR-128-1 and miR-128-2 double knockout mice and inducible miR-128-overexpressing transgenic mice to monitor the proliferation and differentiation of NPCs and their effects on behavior.

Taken together, our results suggest that miR-128 is an important regulator of cortical development through PCM1. Future studies to further elucidate specific aspects of the roles of miR-128 and PCM1 in neuronal development and function will be of great interest to this field.

## Materials and methods

### Animals

All studies were conducted with protocols that were approved by the Institutional Animal Care and Use Committee (IACUC, protocol number: 2013/SHS/809) of the Duke-NUS Graduate Medical

School and National Neuroscience Institute. Time-mated C57BL/6 mice were purchased (InVivos, Singapore) at E13.5 and E14.5 for in utero electroporation and culturing of NPCs.

## Isolation and culture of NPCs

Mouse embryos were harvested at E14.5, and the dorsolateral forebrain was dissected and enzymatically triturated to isolate a population of cells enriched in NPCs as previously described. NPCs isolated from a single brain were suspension-cultured in a T25 tissue culture flask in proliferation medium containing human EGF (10 ng ml$^{-1}$), human FGF2 (20 ng ml$^{-1}$) (Invitrogen, Carlsbad, CA), N2 supplement (1%) (GIBCO), penicillin (100 U ml$^{-1}$), streptomycin (100 mg ml$^{-1}$), and L-glutamine (2 mM) for 5 days and were allowed to proliferate to form neurospheres.

## Transient transfection of NPCs by electroporation

DIV 5 neurospheres were dissociated into single cells using accutase, yielding $4$–$6 \times 10^6$ cells per T25 flask. For each electroporation reaction, $1 \times 10^6$ cells were mixed with 2 µg DNA and electroporated using the Neon electroporator (Invitrogen) according to the manufacturer's protocol. Immediately following electroporation, cells were suspension-cultured in a 6-well tissue culture plate in proliferation medium and were allowed to re-form neurospheres for 24 hr.

## In vitro NPC proliferation assay

Twenty-four hours post-electroporation, the cells were pulsed with 1 mM 5-bromo-2'-deoxyuridine (BrdU, Roche) for 6 hr. The neurospheres were then gently dissociated by pipetting and seeded onto 60 mm coverslips coated with poly-L-lysine and laminin, at a density of $2 \times 10^4$ cells/coverslip. After waiting 30 min to allow the cells to attach, the cells were fixed with 4% paraformaldehyde for 30 min at room temperature. To detect BrdU, NPCs were pre-treated with 2 M HCl for 15 min at 37°C, washed with borate buffer (pH 8.5) for 30 min and immunostained using a mouse antibody against BrdU (#8039, Abcam).

## In vitro NPC differentiation assay

Twenty-four hours post-electroporation, neurospheres were gently dissociated by pipetting and seeded onto 60 mm coverslips coated with poly-L-lysine and laminin, at a density of $2 \times 10^4$ cells/coverslip. Subsequently, NPCs were cultured as monolayer in differentiation medium containing N2 (1%) in DMEM/F12 and were maintained for 5–6 days.

## Immunocytochemistry

The primary antibodies included the following: rabbit anti-Ki67 (#15580, Abcam), mouse anti-beta III tubulin (TUJ1, #1637, Millipore), mouse anti-GFAP(#N206A/8, NeuroMab), mouse anti-NEUN (#MAB377, Millipore), chicken anti-MAP2 (#5392, Abcam), rabbit anti-PH3 (#9701, Cell Signaling), rabbit anti-TBR2 (#23345, Abcam), rabbit anti-cleaved caspase-3 (#9661, Cell Signaling), and rabbit anti-PAX6 (#PRB-278P, Covance). The secondary fluorochrome-conjugated antibodies were diluted 1:400 (donkey anti-mouse, donkey anti-rabbit, goat anti-mouse and goat anti-rabbit, goat anti-chicken (Invitrogen]). Nuclear counterstaining was performed with 4', 6-diamidino-2-phenylindole dihydrochloride (DAPI) (Sigma-Aldrich, #B2261at 0.25 µg/µl). TUNEL staining was carried out using a kit purchased from Roche (#12156792910). Images were obtained using an LSM710 confocal microscope (Zeiss).

## Laser capture microdissection (LCM)

For LCM, unfixed, fresh E14.5 brains were embedded in Tissue-Tek O.C.T. compound (Sakura) at -24°C. Ten-micrometer cryosections were mounted on PEN Membrane slides (Leica Microsystems, Germany) and stored at -80°C until ready for dissection. LCM of the VZ/SVZ, IZ, and CP was performed under direct visualization of the unstained tissue based on tissue morphology using an inverted microscope and PALM Robo software (Carl Zeiss, Germany). The isolated tissue samples were attached onto an Adhesive Cap-500 tube (Carl Zeiss) for direct lysis and total RNA extraction.

### DNA, miRNA, and shRNA constructs

Lentiviral miR-128 expression plasmid and miR-128-Zip knockdown plasmid was obtained from System Biosciences. miR-128 inhibitor was obtained from Genepharma. The PCM1 expression construct was a kind gift from A. Kamiya (Johns Hopkins, US).

To generate luciferase reporter constructs, the 3′-UTR of *Pcm1* was produced by PCR using murine genomic DNA library. Following primers were used: *Pcm1* (1865bp), forward primer: 5′-GAACCTGAAACAGTGGGAGC-3′; reverse primer: 5′-ACGGTTGCATGTTCCCAATC-3′. The resulting PCR products were cloned into the pmirGLO Dual-Luciferase miRNA Target Expression Vector (Promega).

To generate the pmirGLO-PCM1 3′-UTR mutant construct, miR-128 targeting sequences (CACTGTG) were mutated to (GAGTCTG) by the Quick Change Site-Directed Mutagenesis Kit (Stratagene) using following primers: forward primer: 5′-CCTGGACAGATTCAAACCTTGACAGAGTCTGGGATTTTTCTTTTGC-3′ and reverse primer: 5′-GCAAAAGAAAAATCCCAGACTCTGTCAAGGTTTGAATCTGTCCAGG-3′.

We generated two *Pcm1* shRNA vectors with two different published targeting sequences (*Kamiya et al., 2008*) #1: 5′-AGCTACTTAATACAGACTA-3′ and #2: TCAGCTTCGTGATTCTA using pCDH-U6-nucGFP-Puro vector which was modified from the lentiviral miR-128 expression plasmid, MMIR-128-1-PA-1.

### Luciferase assay

HEK293 cells were transfected with either the miR-128 mimic or control miRNA in conjunction with the luciferase reporter constructs. Forty-eight hours after they were transfected, the cells were lysed and subjected to luciferase assays using the Dual Luciferase Reporter Assay System (Promega) according to the manufacturer's protocol.

### RNA isolation and real-time polymerase chain reaction (PCR)

Total RNA was extracted using the miRNeasy kit (Qiagen) from tissue samples or NPCs. The extraction procedure was then followed by cDNA synthesis using a cDNA synthesis kit (Promega). PCRs were performed on three independent sets of template, and the cycling parameters were as follows: 94°C for 15 s, 55°C for 30 s, and 70°C for 30 s for 40 cycles using the CFX96 real-time PCR detection system (Bio-Rad, Hercules, CA). For each assay, PCR was performed after a melting curve analysis. To reduce variability, we ran each sample in duplicate or even triplicate and included control qPCR reactions without template for each run. The qPCR primers are listed in *Figure 4—source data 4*.

### In utero electroporation

Timed-pregnant mice (E13.5) were anesthetized with isoflurane (induction, 3.5%; surgery, 2.5%), and the uterine horns were exposed by laparotomy. The DNA (3–5 µg µl⁻¹ in water) together with the dye Fast Green (2 mg ml⁻¹; Sigma Aldrich, St. Louis, MO) was injected through the uterine wall into one of the lateral ventricles of each embryo using a 30-gauge Hamilton syringe. Approximately 2 µl of DNA and dye solution was delivered using a pressure injector (Picospritzer III; General Valve, Pine Brook, NJ). For electroporation, five electrical pulses (amplitude, 35 V; duration, 50 ms; intervals, 950 ms) were delivered with a BTX square-wave electroporation generator (Harvard Apparatus, Holiston, MA). The uterine horns were placed back into the abdominal cavity after electroporation, and the embryos were allowed to continue their normal development. Electroporated mouse brains were harvested at E14.5 and E17.5 for the proliferation and differentiation analyses, respectively.

### Mitotic spindle orientation assay

The mitotic spindle orientation of dividing cells was determined on micrographs by measuring the angle of chromosomes relative to the ventricular surface. Nuclei were counterstained with DAPI, and mitotic cells were immunostained with PH3. GFP-PH3 double-positive cells within the VZ were identified on micrographs, and approximately 150 cells per experimental condition were used for quantification.

## In vivo NPC proliferation and cell cycle exit assay

For proliferation assays, E14.5 pregnant dams were administered 5-ethynyl-2'-deoxyuridine (5 mg per kg body weight dissolved in 0.9% saline) (EdU, ThermoFisher Scientific) by intraperitoneal injection, and the embryos were harvested 2 hr later. For cell cycle exit assays, E13.5 pregnant dams were IP injected with 5 mg per kg body weight EdU immediately following in utero electroporation, and the embryos were harvested 24 hr later. Visualization of EdU was performed in accordance with the manufacturer's protocol.

## In situ (ISH) and immunohistochemistry

A 5'- digoxigenin (DIG)-labeled locked nucleic acid (LNA) miR-128 ISH detection probe (Exiqon) was used to detect miR-128 expression in the brain. The sequence of the probe was 5'-AAAGAGACCG-GTTCACTGTGA-3'. Briefly, E12.5 to P0 perfused brains were dehydrated in 30% sucrose, embedded with Tissue-Tek, and sectioned at 12 µm (coronal sections) or 14 µm (sagittal sections). Next, the brain slices were treated with 10 µg ml$^{-1}$ Proteinase K at 37°C for 15 min, followed by incubation with the LNA-DIG-labeled miR-128 detection probes (50 nM) at 50°C overnight. The sections were blocked with PBS containing 0.1% Triton X-100, 10% normal goat serum (NGS), and 0.2% bovine serum albumin (BSA) and were incubated with an AP-conjugated anti-DIG antibody (1:1000) in PBS at 4°C overnight. The NBT+CIP substrate was then added (#K2191020, BioChain Institute, Newark, CA). For simultaneous miR-128 ISH and immunostaining, the brain slices were incubated with primary antibodies (peroxidase-conjugated anti-DIG, 1:200 [Roche]; mouse anti-NESTIN, 1:1000 [Sigma]) in Tris-Buffered Saline (TBS) containing 0.1% Tween-20, 20% NGS and 0.1% BSA. Immunostaining was detected using Alexa 488, Alexa 555 or Alexa 647 fluorescent secondary antibodies. Slices were mounted with Vectashield containing DAPI (Vector Labs, Burlingame, CA) and examined with confocal microscopy.

## Confocal image acquisition and quantification

For the analysis of electroporated brain areas, volumetric z-stacks were acquired for each experimental condition using an LSM710 laser confocal microscope and the Z-series images were collapsed with a maximum intensity projection to a 2D representation. To quantify NPC neurogenesis, the Z-series images were taken using a 40x oil immersion objective (NA 1.3) in the CP zone. To quantify the proliferation of NPCs, the Z-series images were taken using a 60x oil immersion objective (NA 1.3) in the SVZ. Approximately 1500–2000 GFP-positive cells were counted for each condition, and at least three sets of independent experiments were performed and manually quantified in a blind manner.

## Immunoblotting

Cells or brain tissues were lysed using RIPA buffer containing phosphatase and protease inhibitors. Proteins were separated by SDS–PAGE under reducing conditions and transferred to polyvinyl difluoride (PVDF) membranes (Millipore, Billerica, MA). Antibody incubations were performed in 5% BSA in TBS buffer using following antibodies: anti-PCM1, 1:1000(Cell Signaling, Danvers, MA); anti-β-actin, 1:1000 (Santa Cruz, Dallas, TX); and HRP-conjugated rabbit or mouse antibodies, 1:3,000 (GE Life Science, Pittsburgh, PA).

## Generation of lentiviral particles

Recombinant lentiviral particles were produced by co-transfection of HEK293T cells with 6 µg of pMD2G-VSVG, 6 µg of PAX2 packaging plasmids, and 0 µg of pCDH-U6-shRNA-nucGFP-Puro plasmid construct using Lipofectamine 2000 (Life Technologies). Lentiviral particles were collected from supernatant after 72 hr by using PEG-it kit (System Biosciences). Lentiviral particles were concentrated up to $2 \times 10^5$ TU/µL.

## Statistical analysis

At least 3 experiments were performed independently under each experiment condition, and similar results were obtained. Statistical analyses were performed using ANOVA and Student's *t*-test. All data were presented as mean and standard deviation (mean ± SD). Statistical significance was defined when ***p<0.001, **p<0.01, *p<0.05 compared with the control.

## Acknowledgements

We are grateful to Dr. Atsushi Kamiya (Johns Hopkins) for mouse PCM1 expression vectors. Also, we thank Drs. Jaewon Ko and Shirish Shenolikar for their critical comments on the manuscript and helpful advice. This work was supported by Singapore Translational Research (STaR) Investigator Award to EKT, A*STAR BMRC TCRP Grant (13/1/96/19/688, to HSJ and LZ), Duke-NUS Signature Research Program Block Grant (to HSJ), and NNI Research Grant (to LZ).

## Additional information

### Funding

| Funder | Grant reference number | Author |
|---|---|---|
| Singapore A*Star | 13/1/96/19/688 | Wei Zhang<br>Hidayat Lokman<br>Hyunsoo Shawn Je<br>Li Zeng |
| Duke-NUS GMS Signature Research Program Grant | | Hyunsoo Shawn Je |
| NNI Research Grant | | Eng King Tan<br>Li Zeng |
| Singapore Translational Research Investigator Award | | Eng King Tan |

The funders had no role in study design, data collection and interpretation, or the decision to submit the work for publication.

### Author contributions

WZ, PJK, Conceived, designed, and performed most of experiments, Conception and design, Acquisition of data, Analysis and interpretation of data, Drafting or revising the article; ZC, HL, LQ, KZ, Performed qPCR analysis, microdissection, and imaging analysis, Acquisition of data, Analysis and interpretation of data; SGR, Performed bioinformatic analysis and edited the paper, Analysis and interpretation of data, Drafting or revising the article; EKT, Provided tissue samples, antibodies, and edited the paper, Drafting or revising the article, Contributed unpublished essential data or reagents; HSJ, LZ, Contributed to ideas, designed, and supervised all experiments, Conception and design, Analysis and interpretation of data, Drafting or revising the article

### Author ORCIDs

Hyunsoo Shawn Je, http://orcid.org/0000-0002-2924-5621

### Ethics

Animal experimentation: All studies were conducted with protocols that were approved by the Institutional Animal Care and Use Committee (IACUC, protocol number: 2013/SHS/809) of the Duke-NUS Graduate Medical School and National Neuroscience Institute.

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
