## [Decision Letter]

[Editors’ note: a previous version of this study was rejected after peer review, but the authors submitted for reconsideration. The first decision letter after peer review is shown below.]

Thank you for choosing to send your work entitled "miR-128 Regulates Proliferation and Neurogenesis of Neural Precursors by Targeting PCM1 in the Developing Neocortex" for consideration at *eLife*. Your full submission has been evaluated by a Senior editor, a Reviewing editor and three peer reviewers, and the decision was reached after discussions between the reviewers. Based on our discussions and the individual reviews below, we regret to inform you that your work will not be considered further for publication in *eLife*.

*Reviewer #1:*

1) The conclusion of the first paragraph is: "qPCR confirmed the observed neural tissue-specific miR-128 expression." But none of the preceding data suggests specificity. For example, "miR-128 was expressed at low levels in the subventricular zone (SVZ) and in the cortical plate (CP) at E14.5…" and "…we performed quantitative real-time PCR (qPCR) and found a similar temporal enrichment of miR-128 in the forebrain, which contains both the SVZ and the CP". Also, "…miR-128 was found to be predominantly expressed in the brain and spinal cord of E14.5 mice…". This does not strike the reviewer as specificity.

2) The authors claim: "High-magnification images of cortical slices at E14.5 revealed the expression of miR-128 in cells within the SVZ and the CP but not within the intermediate zone (IZ) (Figure 1)." The enrichment of the labeling looks convincing for the CP but there is no obvious distinction between the SVZ and the IZ.

3) The isolation procedure of NPCs requires more details regarding its efficiency.

4) The authors state: "The percentage of BrdU- positive but Ki67-negative cells was increased by 350% in NPCs overexpressing miR-128 (Figure 2, arrowheads for BrdU-positive, Ki67-negative cells), indicating that the ectopic expression of miR-128 stimulated NPCs to exit the cell cycle." The conclusion regarding exit from the cell cycle does not follow from the BrdU -positive/Ki67 negative cells.

5) TUNEL is a relatively insensitive way to detect apoptosis and should be complemented by other more conclusive methods.

6) What was the efficiency of the transfection with CMV-promoter-driven expression of the mouse miR-128-1 precursor and EF1α promoter-driven copGFP constructs?

7) The claim of an effect on neuronal differentiation is a good start, but their study requires more detail regarding exactly what the effects are on differentiation and on specific cell types. The paper as it now stands reads like an isolated observation without larger conceptual insights arising from the data presented.

*Reviewer #2:*

The development of the cerebral cortex involves a tight coordination of progenitor proliferation, neuron migration and differentiation. The present work analyses the function of miR-128 during cerebral cortex development, which is a microRNA misregulated in autism spectrum disorder. The findings suggest that miR-128 is expressed by cortical progenitors where it controls their proliferation and differentiation by repressing the expression of PCM1, a protein previously described as critical for cell-cycle regulation. Ectopic expression of miR-128 promotes cell cycle exit and neurogenesis in vitro as well as in the cortex in vivo. This phenotype is rescued by PCM1 gain of function and mimicked by its loss expression, suggesting that PCM1 is a downstream target of miR-128 critical for cortical neurogenesis. Thus miR128 may belong to the machinery that coordinate cell cycle exit with neuronal differentiation.

1) According to its marked detection in the CP but not in VZ/SVZ, miR-128 has recently been shown to be solely important for late aspects of corticogenesis (migration and differentiation of projection neurons) (Franzoni et al., 2015). The results of the present findings suggest an early function of miR-128 and the reviewer highly recommends clarification of the discrepancy observed between both works.

2) The authors claim that miR-128 is not expressed in the IZ. The low magnification in Figure 1 is not conclusive and doesn't allow the reviewer to take a conclusion. One suitable experiment is qRT-PCR analysis of miR-128 and its corresponding pre-miR128 on laser-captured cortical wall regions.

3) The described gain of function phenotypes in vitro (Figure 2) is incomplete and should include more information about the experimental procedures in the text (e.g. length of BrdU pulses for cell cycle exit, timing of cultures, etc.).

4) Additional loss of function experiments (using antagomiRs or sponges against endogenous miR-128) should be done to support a physiological function of miR-128 in NPCs. This remark also stands for in utero (in vivo) experiments.

5) The monitoring of AP (=RGC) and IP cell populations using Pax6 and Tbr2, respectively is inconclusive. Other markers should be checked (such as Sox2 for AP and Insm1 for IP) to exclude specific marker loss after miR-128 expression rather than progenitor population loss.

6) The authors claim that overexpression of miR-128 "accelerates" cell cycle exit in vivo, however this is not experimentally demonstrated. In addition, promoting the generation of IPs is not really matching with increase cell cycle exit as these progenitors retain the ability to cycle and generate the bulk of projection neurons. This should be clarified and rephrased in the text.

7) It is not clear what parameter overexpression of miR-128 is affecting in APs. Is it only promoting the generation of IP or also inducing direct neurogenesis? This should be experimentally addressed. Moreover, these experiments should be complemented by loss of function studies as mentioned above. In its present form, the manuscript does not address any physiological roles of mir-128 in corticogenesis.

8) While there is no doubt that miR-128 can target PCM1 mRNAs in vitro, the expression of PCM and its targeting by endogenous miR-128 should be assessed in vivo (immunolabelings, etc.). Along this line the results obtained with functional experiments performed in vitro with miR-128 or PCM1 (proliferation and differentiation) should be confirmed in vivo. This is important because cultured NPC are different from resident APs and IPs (gene expression pattern deregulated to some extent).

*Reviewer #3:*

In this manuscript, the authors investigated the function of miR-128 in cortical neural differentiation. They showed that miR-128 enhances neuronal differentiation and represses proliferation of NPCs through PCM1. The results are interesting. However the data are limited and impact is moderate. I have several major concerns.

First, the experiments are not complete. For gene or miRNA functional assays, both gain and loss of function assays should be performed, but not. The justification of ASD link is based on a previous study showing that miR-128 is one of 28 microRNAs dysregulated in at least one ASD individuals (for miR-128, the level is decreased in one ASD) (Abu-Elneel Neurogenetics 2008). However, several experiments only show the impact of overexpression of miR-128, not reduction, on NPCs.

Second, the in vitro NPC assays were done using electroporation which introduces a large amount of genetic materials into the cells. Electroporation also leads to a lot of cell death which is well known. However, these are not addressed. These experiments should be done, or at least confirmed using lentiviral infection.

Third, the paper is not well prepared. There is a lack of details both in methods and in legends.

Fourth, many of the experiments do not have appropriate controls.

Fifth, the statistical analyses were not done correctly.

[Editors’ note: what now follows is the decision letter after the authors submitted for further consideration.]

Thank you for resubmitting your work entitled "MiRNA-128 Regulates the Proliferation and Neurogenesis of Neural Precursors by Targeting PCM1 in the Developing cortex" for further consideration at *eLife*. Your revised article has been favorably evaluated by a Senior editor, a Reviewing editor, and three reviewers. The manuscript has been improved but there are some remaining issues that need to be addressed before acceptance, as outlined below:

Briefly, the reviewers pointed out some misinterpretations, unclear descriptions, and technical shortcomings, and suggested to perform some more experiments to clarify these issues. More importantly, the reviewers raised a concern about the lack of the data from knockout mice, although I am well aware of that you already have in utero electroporation loss-of-function data (semi-in vivo data) in the manuscript. These data may take a while to obtain, but you may already be close to these data or can consider using the Crispr technology to speed up the processes. Inclusion of these data, in addition to the above mentioned revision experiments, would much strengthen the manuscript.

*Reviewer #1:*

In this revised application, the authors have provided a substantial amount of data. In fact, the authors have provided new data to address all of my concerns, except for one, which is evidence for in vivo targeting of PCM1 by miR-128. However, given the substantial other evidence supporting the regulation of PCM1 by miR-128, I consider this is an optional data at this point. I suggest acceptance for publication.

*Reviewer #2:*

1) They note that an early expansion of NPC pools by miR-128 knockdown did not result in a net increase in neuronal numbers. That is a curious result and their explanation regarding a critical time window does not seem like an adequate explanation.

2) They did a number of experiments to validate PCM1 as a target miR-128, but they failed to do one of the most important experiments, i.e. show that the "anti-miR" can increase the endogenous protein. They show that miR-128 over expression can reduce the protein, but over-expression experiments are never as strong as the knock down experiment.

3) They need to make clearer how they chose genes for qPCR. Did they look at every gene in their list of predicted targets? Which list? How was the list filtered? How did they settle on PCM1 among all the genes affected by over-expression? Was it the only one also affected by the miR-128 inhibitor? What is the role of the other genes on the list that are potential miR-128 targets?

4) Did PCM1 knock down or over-expression affect spindle orientation?

5) miRNAs generally have small effects-they tend to buffer cells to assure a precision outcome. This paper would lead one to think that miR-128 has effects in development that go far beyond what is likely to be the in vivo role of this miRNA. The system they are analyzing which controls neural precursor numbers and spindle orientation in precursors is likely to be far more complex than what is presented here, and hence the paper seems like a simplification of a much deeper biological question.

*Reviewer #3:*

The updated version of the manuscript includes new set of data that answer several of my concerns. However, I have additional comments about the manuscript in its present form.

1) One major issue concerns the interpretation of the data obtained in vitro with BrdU incubation. According to their clarified BrdU experiment procedure. I noticed a misinterpretation of the data. Indeed, conversely to its fast clearance from brain tissue in vivo, the BrdU remains pretty stable (for several hours) in the culture medium and it is necessary to replace it after a short time to properly assess a single cohort of cells undergoing S-phase. This is an issue when it comes to monitor cell cycle exit rate of such cohort (Figure 2–Figure 6). In these bioassays, the cell cycle rate exit is totally biased by the continuous labelling (during 24 hours or so) of proliferative cells undergoing S-phase (which is indeed different between conditions). Same comments stand when assessing S-phase in proliferation assays with 6hours of incubation in BrdU-containing medium.

2) The conclusion about mitotic spindle orientation (lanes 169-17) is incorrect. The Figure 3 indeed show that acute modulation of miR128 expression affect spindle orientation. It is not possible to conclude that default of mitotic spindle orientation leads to impaired APs division (symmetric vs asymmetric). This has been discussed in several recent publications. A clonal analysis would be required to follow the fate of the AP progenies after miR-128 modulation. One complementary set of experiments to decipher the functional impact of mitotic spindle orientation modification would be to focally electroporate APs (with miR-128 or miR-Zip-128) and identify the fate of the progenies 24 hours later (Tbr1 for neuron, Pax6 for APs and Tbr2 for IPs).

[Editors' note: further revisions were requested prior to acceptance, as described below.]

Thank you for submitting your work entitled "MiRNA-128 Regulates the Proliferation and Neurogenesis of Neural Precursors by Targeting PCM1 in the Developing cortex" for consideration by *eLife*. Your article has been reviewed by two peer reviewers, and the evaluation has been overseen by a Reviewing Editor and a Senior Editor.

The reviewers have discussed the reviews with one another and the Reviewing Editor has drafted this decision to help you prepare a revised submission.

The current version of the manuscript with additional data has significantly improved, compared with the previous version.

However, there is still one remaining point that should be fixed. Like mentioned previously, keeping BrdU in NPC culture (which is more stable than in vivo) to assess proliferation is OK if the authors do not refer to a single cohort of cells that have undergone S-phase (otherwise the authors should have only treated cells for about an hour with BrdU).

Now, concerning the cell cycle exit rate, we usually monitor a single cohort of cell that has exited the cell cycle. It is thus not possible to measure it properly if the BrdU is kept during 24 hours in vitro (stable) because several cohorts of cells will undergo S-phase at different timing, further leading to biased analyses. However, the cell cycle exit is properly addressed in vivo (where clearance of BrdU is fast). I would thus suggest to get rid of the in vitro data that are poorly executed.

---

## [Author Response]

[Editors’ note: the author responses to the first round of peer review follow.]

Upon receiving the reviewers’ comments and suggestions, we carefully examined our manuscripts and found that none of the reviewers pointed out lack of novelty in our study or conceptual and experimental flaws. One of the reviewers found our study is rather “interesting” and all reviewers suggested better clarification of methodologies employed and demanded for new data by performing additional experiments. Therefore, to address the concerns raised by reviewers, we have performed the requested additional experiments and revised our manuscript, which now includes extensive new data and clarifications. Our specific, point-by-point responses to the reviewers are listed below:

• We have included new data showing the effect of knocking down miR-128 using an anti- sense short hairpin targeted at miR-128 (miR-ZIP-128) both in vitro NPC culture, and in vivo, following in utero electroporation. We found that loss-of-function in miR-128 has the opposite effect on neurogenesis in vitro (Figure 2) and in vivo (Figure 3).

• We have included new data demonstrating mitotic spindle orientation changes following miR-128 overexpression and knockdown in vivo (Figure 3). We found that miR-128 overexpression increase neurogenesis by enhancing indirect neurogenesis from basal neuronal progenitor cells

• We have repeated the cell proliferation experiments in vivo using EdU to supplement our previous findings with PH3 and Ki67. We obtained in vivo results that were consistent with our previous in vitro results (Figure 3; Figure 3).

• We have included new qPCR data using tissue samples derived from laser capture microdissection of E14.5 mouse brain, showing the levels of miR-128 expression in distinct tissue regions (Figure 4).

• We have repeated the in vitro NPC experiments with lentivirus-mediated gene delivery of miR-128 and miR-ZIP-128. We demonstrated that, compared to the electroporation method, the transfection efficiency is improved, although the overall effect on neurogenesis remains unchanged.

• We have repeated the miR-128 target gene analysis to increase statistical power. We have included the results in the revised manuscript.

• We have revised the Materials and methods section as requested.

• We have revised the results and discussion sections as requested.

*Reviewer #1:*

Reviewer #1 recommended to perform additional experiments and to clarify the interpretation of the data as well as experimental procedures. We thank the reviewer for his/her comments and have addressed concerns as follows:

*1) The conclusion of the first paragraph is: "qPCR confirmed the observed neural tissue-specific miR-128 expression." But none of the preceding data suggests specificity. For example, "miR-128 was expressed at low levels in the subventricular zone (SVZ) and in the cortical plate (CP) at E14.5…" and "…we performed quantitative real-time PCR (qPCR) and found a similar temporal enrichment of miR-128 in the forebrain, which contains both the SVZ and the CP". Also, "…miR-128 was found to be predominantly expressed in the brain and spinal cord of E14.5 mice…". This does not strike the reviewer as specificity.*

The reviewer is correct here. We have edited the main text (Results).

*2) The authors claim: "High-magnification images of cortical slices at E14.5 revealed the expression of miR-128 in cells within the SVZ and the CP but not within the intermediate zone (IZ) (Figure 1)." The enrichment of the labeling looks convincing for the CP but there is no obvious distinction between the SVZ and the IZ.*

To address this, we performed laser capture micro-dissection to isolate cells within three brain regions (VZ/SVZ, IZ, CP) and measured the levels of endogenous miR-128 expression using qPCR. We confirmed that endogenous miR-128 is enriched within the CP region, and there is no difference between the VZ/SVZ and IZ (Figure 4) In addition to confirm miR-128 expression in the VZ/SVZ, we have included new ISH and IHC co-localization data to show that miR-128 is expressed in nestin-positive NPCs within the VZ/SVZ at E14.5 (Figure 1—figure supplement 1). We have edited the main text to reflect these new findings (subsection “miR-128 is expressed in NPCs in the developing murine cortex”).

*3) The isolation procedure of NPCs requires more details regarding its efficiency.*

We have revised the description of the isolation procedures to include the number of dissected cortices, the number of NPCs following suspension culture, and numbers of NPCs used for electroporation (Materials and methods). We have also included schematic representations to illustrate the in vitro cell proliferation, cell cycle exit and neuronal differentiation experimental procedures (Figure 2).

*4) The authors state: "The percentage of BrdU-positive but Ki67-negative cells was increased by 350% in NPCs overexpressing miR-128 (Figure 2, arrowheads for BrdU-positive, Ki67-negative cells), indicating that the ectopic expression of miR-128 stimulated NPCs to exit the cell cycle." The conclusion regarding exit from the cell cycle does not follow from the BrdU -positive/Ki67 negative cells.*

We agree that the statement in the original manuscript was not clear and we would like to clarify our cell cycle exit assay. First, all NPCs that have undergone S-phase of the cell cycle were labelled with BrdU by pulse labelling the electroporated cells for 24 hours. Then we used Ki67 (labels G1, S, G2 and M-phase) as a negative marker for cells that have exited the cell cycle. In short, NPCs that have exited the cell cycle following S-phase were identified as cells that were positive for GFP and BrdU, but negative for Ki67. Our results showed that the percentage of cells that exited the cell cycle increased 350% in NPCs overexpressing miR-128 compared to miR-mimic (Figure 2). To rule out cell death as a cause we performed both TUNEL assay and antibody staining against cleaved caspase-3 to show that miR-128 did not affect apoptosis (Figure 2—figure supplement 2). Taken together, we concluded that ectopic expression of miR-128 induced cell cycle exit in NPCs.

*5) TUNEL is a relatively insensitive way to detect apoptosis and should be complemented by other more conclusive methods.*

We agree with the reviewer and would like to clarify that we had already performed immunostaining to analyze cleaved caspase-3, a marker for apoptosis, in our original manuscript (subsection “miR-128 regulates the proliferation and differentiation of NPCs in vitro”, third paragraph) (Figure 2—figure supplement 2).

*6) What was the efficiency of the transfection with CMV-promoter-driven expression of the mouse miR-128-1 precursor and EF1α promoter-driven copGFP constructs?*

We used a dual-promoter expression construct featuring a CMV-promoter that drives the expression of miR-128 precursor and an EF1α promoter that drives copGFP upon expression of a single plasmid vector. The transfection efficiency for the two elements (miR-128 and copGFP) will always be the same. We have edited the main text (subsection “miR-128 regulates the proliferation and differentiation of NPCs in vitro”) and included graphical representation of the constructs to highlight the dual promoter arrangement (Figure 2—figure supplement 1).

Figure 8 illustrates the transfection efficiency following electroporation of NPCs in vitro, using aforementioned constructs, which indicate 50% of cells express GFP after 24 hours.

Author response image 1.Transfection efficiency of expression constucts.NPCs were electroporated with the indicated plasmids and the nuclei were counterstained with DAPI. White outlines indicate GFP_+_ cells and red outlines indicate cells have that borderline expression of GFP and considered GFP-negative. White arrow indicates a GFP-negative cell. Quantification of the ratio of GFP-positive cells over total DAPI counterstained cells show the transfection efficiency to be consistent between the miR-128 overexpression construct and its corresponding control. More than 1500 GFP-positive cells were counted for each condition. At least three sets of independent experiments were performed. The values represent the means ± s.d. (n=3). Student’s *t*-test, differences were considered significant at ****P<0.001*.**DOI:**
http://dx.doi.org/10.7554/eLife.11324.035

*7) The claim of an effect on neuronal differentiation is a good start, but their study requires more detail regarding exactly what the effects are on differentiation and on specific cell types. The paper as it now stands reads like an isolated observation without larger conceptual insights arising from the data presented.*

We respectfully disagree with the reviewer. In our manuscript, we present experimental evidence suggesting that miR-128 regulates self-renewal and neuronal differentiation of NPCs by targeting PCM1 during early cortical development in a rigorous manner. We further strengthened this evidence by performing experiments suggested by the reviewers. For example: 1) We have included new qPCR data using tissue samples from laser capture microdissection of E14.5 mouse brain, showing the levels of miR-128 expression in distinct tissue regions; 2) We have included new data showing the effect of knocking down miR-128 using an anti-sense short hairpin targeted at miR-128 both in vitro NPC culture, and in vivo, following in utero electroporation; 3) We have included new data demonstrating spindle orientation changes following miR-128 overexpression and knockdown in vivo. We believe that our findings greatly enhance the understanding of the functional role that miR-128 plays in the homeostasis of neural stem cells through its multiple layers of exquisite control during early cortical development.

*Reviewer #2: The development of the cerebral cortex involves a tight coordination of progenitor proliferation, neuron migration and differentiation. The present work analyses the function of miR-128 during cerebral cortex development, which is a microRNA misregulated in autism spectrum disorder. The findings suggest that miR-128 is expressed by cortical progenitors where it controls their proliferation and differentiation by repressing the expression of PCM1, a protein previously described as critical for cell-cycle regulation. Ectopic expression of miR-128 promotes cell cycle exit and neurogenesis in vitro as well as in the cortex in vivo. This phenotype is rescued by PCM1 gain of function and mimicked by its loss expression, suggesting that PCM1 is a downstream target of miR-128 critical for cortical neurogenesis. Thus miR128 may belong to the machinery that coordinate cell cycle exit with neuronal differentiation.*

Reviewer #2 suggested clarifying the discrepancy between Franzoni et al. and ours on the role of miR-128 in corticogenesis. In addition, he/she suggested performing additional loss-of-function experiments to support a physiological his/her comments and have addressed the concerns as follows:

*1) According to its marked detection in the CP but not in VZ/SVZ, miR-128 has recently been shown to be solely important for late aspects of corticogenesis (migration and differentiation of projection neurons) (Franzoni et al., 2015). The results of the present findings suggest an early function of miR-128 and the reviewer highly recommends clarification of the discrepancy observed between both works.*

Although we observed that miR-128 is highly enriched in the CP, suggesting that miR-128 could play a role in late corticogenesis, we respectfully disagree with the reviewer’s view that miR-128 is solely important for late aspects of corticogenesis. The paper by Franzoni et al. provided insufficient evidence to rule out potential involvement of miR-128 during early corticogenesis, as most of their results were obtained upon miR-128 overexpression during late corticogenesis, given that they performed in uteroelectroporation at E15.5, at which early neurogenesis has largely concluded (Franzoni E, Booker SA, Parthasarathy, 2015. Furthermore, they did not perform any analysis on neural progenitor proliferation and renewal, which was not the focus of their study.

In addition, as mentioned in our response to reviewer #1’s comments (see point 2 above) we have included new ISH and IHC co-localization data to show that miR-128 is expressed in nestin positive NPCs within VZ/SVZ at E14.5 using LNA probes (Figure 1—figure supplement 1). Furthermore, Ge et al. demonstrated that PCM1 knockdown in NPCs leads to-1) decreased NPC proliferation, 2) increased cell-cycle exit of NPCs, and 3) overproduction of neurons (Ge et al., 2010), which were in accordance with our miR-128 overexpression data during early cortical neurogenesis. It will be interesting to examine the function of Phf6 as a potential downstream target of miR-128 during early neurogenesis, but we thought this would be out of the scope for our current manuscript. Taken together, we argue that there is little overlap or discrepancy between Franzoni et al. and our results. We have included these points in our revised Discussion (third paragraph).

*2) The authors claim that miR-128 is not expressed in the IZ. The low magnification in Figure 1 is not conclusive and doesn't allow the reviewer to take a conclusion. One suitable experiment is qRT-PCR analysis of miR-128 and its corresponding pre-miR128 on laser-captured cortical wall regions.*

As mentioned in our response to reviewer #1’s comments (see point 2 above) we performed laser-capture micro-dissection to isolate NPCs from the three regions (VZ/SV, IZ, and CP) and analysed the expression levels of miR-128 using qPCR (Figure 4). We confirmed that endogenous miR-128 is enriched within the CP region, and there is no difference between the VZ/SVZ and IZ. We have edited the main text to reflect these new findings.

*3) The described gain of function phenotypes in vitro (Figure 2) is incomplete and should include more information about the experimental procedures in the text (e.g. length of BrdU pulses for cell cycle exit, timing of cultures, etc.).*

As mentioned in our response to reviewer #1’s comments (see point 2 above) we have revised the description of the isolation procedures to include the number of dissected cortices, the number of NPCs following suspension culture, and numbers of NPCs used for electroporation Materials and methods). We have also included schematic representations to illustrate the in vitro cell proliferation, cell cycle exit and neuronal differentiation experimental procedures (Figure 2).

*4) Additional loss of function experiments (using antagomiRs or sponges against endogenous miR-128) should be done to support a physiological function of miR-128 in NPCs. This remark also stands for in utero (in vivo) experiments.*

We agreed with the reviewer and performed the loss-of-function experiment using a construct that expresses anti-sense miR-128 short-hairpin RNAs (miR-Zip-128)(Guibinga et al., 2012). The passages below include excerpts from our updated manuscript (Results), modified to highlight the findings from the miR-128 loss-of-function experiments:

First, expression of miR-Zip-128 in cultured NPCs increased BrdU incorporation, but reduced the number of cells that exit from the cell cycle, which are indicative of a net increase in proliferative capacity of NPCs. Furthermore, miR-Zip-128 expression in NPCs decreased neural differentiation, as measured by Tuj1 staining (Figure 2).

Then, we performed miR-128 loss-of-function experiments in vivoby in uteroelectroporation of miR-Zip-128 at E13.5. First we monitored mitotic spindle orientation in dividing apical progenitors (AP) within the VZ/SVZ and found that miR-128 knockdown led to a significant increase in horizontally dividing cells (Figure 3).

The observation was further supported when we found that miR-128 knockdown cells showed a significant increase in EdU incorporation (Figure 3), a significant increase in Ki67-positive cells (Figure 3—figure supplement 1) and a significantly higher percentage of cells positive for the AP markers PAX6 (Figure 3) and SOX2 (Figure 3—figure supplement 2), indicating an increase in actively proliferating APs in the VZ/SVZ.

Next, we examined whether the significant decrease in obliquely dividing cells following miR-128 knockdown was indicative of an eventual contraction of basal progenitors (BP) by using TBR2 as a marker. miR-128 knockdown led to a significant decrease in the number of TBR2-positive cells (Figure 3). Taken together we concluded that miR-128 knockdown regulates the NPC population by promoting AP proliferation while inhibiting BP generation.

As the BPs are responsible for the bulk of cortico-neurogenesis (Tan and Shi, 2013), we further asked whether miR-128 affects NPC differentiation. First we analysed the change in the number of BPs exiting the cell cycle following miR-128 knockdown. We found that compared to control, miR-128 knockdown cells showed a significant decrease in cells that were GFP- and Edu-positive but Ki67-negative indicating decreased cell cycle exit.

Finally we asked whether the observed changes in cell cycle exit ultimately led to a difference in neurogenesis. We analyzed brains at E17.5, four days after electroporation and neuronal differentiation was assayed by immunostaining with a specific antibody against the neuronal marker NeuN. Intriguingly, miR-128 knockdown significantly decreased the number of NeuN-positive neurons in the CP compared to control (Figure 3).

Taken together, suppression of endogenous miR-128 in early neural precursors impedes their developmental progression by retaining NPCs in a more primitive and proliferative state. Most intriguingly, this early expansion of NPCs did not translate into a net increase in neuronal numbers, suggesting that there is a potential critical time window in which the progenitors must switch to a differentiation state, otherwise, terminally compromising their neurogenic capacity.

*5) The monitoring of AP (=RGC) and IP cell populations using Pax6 and Tbr2, respectively is inconclusive. Other markers should be checked (such as Sox2 for AP and Insm1 for IP) to exclude specific marker loss after miR-128 expression rather than progenitor population loss.*

We performed additional experiments to monitor SOX2 expression, which was consistent with our previous findings (Figure 3—figure supplement 2). Although we didn’t monitor IP cell populations, we performed alternative experiments to monitor mitotic spindle orientation in dividing APs within the VZ. In this experiment, we demonstrated that miR-128 expression led to a significant decrease in horizontally dividing cells, which are in accordance with a contraction of the AP cell population, but a significant increase in obliquely dividing cells, which indicated an expansion of the IP cell population (Huttner and Kosodo, 2005; Wang, Lui and Kriegstein, 2011). In addition, expression of miR-Zip-128 resulted in the opposite effect from miR-128 expression. We have included these new findings in our revised manuscript (subsection “miR-128 promotes neuronal differentiation in the developing cortex in vivo”) (Figure 3).

*6) The authors claim that overexpression of miR-128 "accelerates" cell cycle exit in vivo, however this is not experimentally demonstrated. In addition, promoting the generation of IPs is not really matching with increase cell cycle exit as these progenitors retain the ability to cycle and generate the bulk of projection neurons. This should be clarified and rephrased in the text.*

Regarding the first point mentioned, we agree with the reviewer that our data does not suggest an “acceleration” of cell cycle exit. We have rephrased our conclusion to highlight the final increase in BrdU-positive/Ki67-negative cells after miR-128 overexpression (Figure 3). In regards to the second point, the increase in cell cycle exit is observed in the basal progenitors or IPs within the IZ. The increased cell cycle exit of IPs is consistent with our other findings that miR-128 overexpression leads to increase neurogenesis in vivo (Figure 3). We have included these points and revised our manuscript accordingly (subsection “miR-128 promotes neuronal differentiation in the developing cortex in vivo”).

*7) It is not clear what parameter overexpression of miR-128 is affecting in APs. Is it only promoting the generation of IP or also inducing direct neurogenesis? This should be experimentally addressed. Moreover, these experiments should be complemented by loss of function studies as mentioned above. In its present form, the manuscript does not address any physiological roles of mir-128 in corticogenesis.*

As mentioned in our response to the comments above, mitotic spindle orientation of APs in the VZ/SVZ suggests that miR-128 overexpression does not affect direct neurogenesis as reflected by no change in vertically (60-90°) dividing cells (Shitamukai and Matsuzaki, 2012) (Figure 3).

*8) While there is no doubt that miR-128 can target PCM1 mRNAs in vitro, the expression of PCM and its targeting by endogenous miR-128 should be assessed in vivo (immunolabelings, etc.). Along this line the results obtained with functional experiments performed in vitro with miR-128 or PCM1 (proliferation and differentiation) should be confirmed in vivo. This is important because cultured NPC are different from resident APs and IPs (gene expression pattern deregulated to some extent).*

Regarding the first point, we performed two sets of experiments to determine if PCM1 is an endogenous in vivotarget of miR-128. First, the qPCR analysis showed that temporal expression of miR-128 was gradually increased in the forebrain of the embryo at E12.5, 14.5, 17.5, and P0 (Figure 4—figure supplement 2). In contrast, PCM1 protein level was found to gradually decrease in the forebrain at these time points (Figure 4—figure supplement 2). Next, LCM (revealed that miR-128 was highly expressed in the CP zone of the mouse cerebral cortex by qPCR, compared to IZ and SVZ/VZ. Conversely, qPCR analysis showed PCM1 was decreased in the CP zone compared to IZ and SVZ/VZ (Figure 4). These results (as well our bioinformatics and expression analyses of potential targets) suggest that PCM1 can be an in vivotarget of miR-128. We have included these new findings and revised our manuscripts (subsection “PCM1 is a direct target of miR-128 in NPCs”).

In regards to the second point, we agree with the reviewer that cultured NPCs could be different from those neural progenitors in vivo. With that in mind, we have originally designed the experiments to study the functional changes resulting from miR-128 in NPC culture as an in vitromodel, followed by in vivoexperiments using in uteroelectroporation of embryonic brain. To reflect this original intention, we have extensively revised the manuscript by rearranging figures and adding more data.

*Reviewer #3: In this manuscript, the authors investigated the function of miR-128 in cortical neural differentiation. They showed that miR-128 enhances neuronal differentiation and represses proliferation of NPCs through PCM1. The results are interesting. However the data are limited and impact is moderate. I have several major concerns.*

Reviewer #3 focuses on function of miR-128 in cortical neural differentiation. They showed that miR-128 enhances neuronal differentiation and represses proliferation of NPCs through PCM1. The results are interesting. However the data are limited and impact is moderate. We thank the reviewer for his/her comments and have addressed the concerns raised as follows:

*First, the experiments are not complete. For gene or miRNA functional assays, both gain and loss of function assays should be performed, but not. The justification of ASD link is based on a previous study showing that miR-128 is one of 28 microRNAs dysregulated in at least one ASD individuals (for miR-128, the level is decreased in one ASD) (Abu-Elneel Neurogenetics 2008). However, several experiments only show the impact of overexpression of miR-128, not reduction, on NPCs.*

We thank the reviewer for the valuable comment. As mentioned in our response to reviewer #2’s comments (see point 4 above) we used a construct that expresses anti-sense miR-128 short hairpin RNAs (miR-Zip-128) and performed miR-128 loss-of-function both in vitroand in vivo. The new findings have been included in the updated manuscript (Results) (Figure 2, Figure 3).

*Second, the in vitro NPC assays were done using electroporation which introduces a large amount of genetic materials into the cells. Electroporation also leads to a lot of cell death which is well known. However, these are not addressed. These experiments should be done, or at least confirmed using lentiviral infection.*

We thank the reviewer for the valuable comment. We have repeated the in vitro NPC experiments replacing electroporation with lentivirus vectors to deliver the miR-128 overexpression and knockdown constructs. We noted that the efficiency of transfection is improved whereas the overall effect remains unchanged. We have included these points in the updated manuscript (subsection “miR-128 regulates the proliferation and differentiation of NPCs in vitro” and Figure 2—figure supplement 4).

Third, the paper is not well prepared. There is a lack of details both in methods and in legends.

We thank the reviewer for the valuable comment. We have included further details in the Materials and methods section and revised the figure legends.

*Fourth, many of the experiments do not have appropriate controls.*

*Fifth, the statistical analyses were not done correctly.*

We thank the reviewer for the valuable comment. We have made the appropriate changes in the manuscript. As mentioned below we have updated our manuscript and used one-way ANOVA to test our hypotheses where appropriate.

[Editors' note: the author responses to the re-review follow.]

*Briefly, the reviewers pointed out some misinterpretations, unclear descriptions, and technical shortcomings, and suggested to perform some more experiments to clarify these issues. More importantly, the reviewers raised a concern about the lack of the data from knockout mice, although I am well aware of that you already have in utero electroporation loss-of-function data (semi-in vivo data) in the manuscript. These data may take a while to obtain, but you may already be close to these data or can consider using the Crispr technology to speed up the processes. Inclusion of these data, in addition to the above mentioned revision experiments, would much strengthen the manuscript. Reviewer #2: 1) They note that an early expansion of NPC pools by miR-128 knockdown did not result in a net increase in neuronal numbers. That is a curious result and their explanation regarding a critical time window does not seem like an adequate explanation.*

We agreed with the reviewer that our explanation in the previous manuscript was merely speculation and might not be an adequate explanation for the phenomenon. Therefore, we have made changes in our revised manuscript accordingly (subsection “miR-128 promotes neuronal differentiation in the developing cortex in vivo“, fifth paragraph).

*2) They did a number of experiments to validate PCM1 as a target miR-128, but they failed to do one of the most important experiments, i.e. show that the "anti-miR" can increase the endogenous protein. They show that miR-128 over expression can reduce the protein, but over-expression experiments are never as strong as the knock down experiment.*

We agree with the reviewer that knock down experiments are crucial for validating the relationship between miR-128 and its proposed target, PCM1. That was the reason why we performed a knock down experiment using anti-miR-128 in primary mouse NPCs and detected change in endogenous PCM1 protein level via western blot analysis in our original manuscript (Figure 4). Furthermore, we used the luciferase assay to validate the binding specificity of miR-128 with its predicted target, the 3-UTR of the *Pcm1* gene. Upon confirmation of the binding, we performed separate experiments to monitor endogenous protein level change in NPCs by using either miR-128 overexpression or knockdown in mouse NPCs (Figure 4).

*3) They need to make clearer how they chose genes for qPCR. Did they look at every gene in their list of predicted targets? Which list? How was the list filtered? How did they settle on PCM1 among all the genes affected by over-expression? Was it the only one also affected by the miR-128 inhibitor? What is the role of the other genes on the list that are potential miR-128 targets?*

We have prepared the diagram below (Figure 9) to clarify the processes in identifying miR-128 target genes. In addition, we have modified the revised manuscript to better describe rationale of choosing PCM1 as the putative target of miR-128 based on its functional role (subsection “miR-128 promotes neuronal differentiation in the developing cortex in vivo“).

Author response image 2.Schematic diagram outlines rationale of gene selection process.Two widely used in silico microRNA target prediction databases (TargetScan and miRanda)(Mi et al., 2013; Witkos et al., 2011) predicted 800 and 940 potential targets of miR-128, respectively. Among the 77 overlapping targets only 53 were annotated to have known biological function and only 11 out of the 53 genes exhibited consistent reduction in mRNA levels upon miR-128 overexpressionin cultured mouse NPCs and were further tested for reciprocal upregulation when miR-128 was inhibited. Lastly, upon miR-128 inhibition in NPCs, only *Pcm1, Nfia, Foxo4*, and *Fbxl20* were consistently upregulated.**DOI:**
http://dx.doi.org/10.7554/eLife.11324.036

Briefly, among four candidate targets identified from multiple round of validation, *Foxo4*, which encodes for an insulin/IGF-1 responsive transcription factor that regulates cell cycles (Furukawa-Hibi et al., 2005; Schmidt et al., 2002), was ruled out because a recent study reported that the loss of FOXO4 reduced the potential of human embryonic stem cells to differentiate into neural lineages(Vilchez et al., 2013), which was opposite from the cellular phenotype of NPCs upon miR-128 overexpression. *Nfia* and *Fbox120* were ruled out, because *Nfia* protein functions as a transcription and replication factor for adenovirus DNA replication (Qian et al., 1995) and Fbx120 protein is involved in synaptic plasticity, which occurs postnatally (Takagi et al., 2012). Therefore, we sought to focus on PCM1 as our primary gene of interest (Figure 4–Figure 7).

*4) Did PCM1 knock down or over-expression affect spindle orientation?*

We carried out the spindle orientation assay following PCM1 overexpression using in utero electroporation in E13.5 mice. The result was still similar to that of miR-128 knockdown (original manuscript, Figure 3). We observed an increase in percentage of horizontally dividing cells and a decrease in percentage of obliquely dividing cells, which is consistent with our previous observations of increased numbers of PAX2-positive cells and decreased numbers of TBR2-positive cells upon PCM1 overexpression (original manuscript, Figure 5—figure supplement 7). We included these data in Figure 5—figure supplement 7, and described these results in the revised manuscript (subsection “PCM1 regulates the proliferation and differentiation of NPCs”, last paragraph).

*5) miRNAs generally have small effects-they tend to buffer cells to assure a precision outcome. This paper would lead one to think that miR-128 has effects in development that go far beyond what is likely to be the in vivo role of this miRNA. The system they are analyzing which controls neural precursor numbers and spindle orientation in precursors is likely to be far more complex than what is presented here, and hence the paper seems like a simplification of a much deeper biological question.*

We agree that miRNAs tend to function as fine-tuning modulators of cellular processes(Bartel, 2009; Kawahara et al., 2012; Li and Jin, 2010; Shi et al., 2010). However, there are numerous examples of how miRNAs exert significant functional changes upon perturbation(Franzoni et al., 2015; Peruzzi et al., 2013; Tan et al., 2013). In addition, while we are currently obtaining miR-128-1 and -2 double knockout mice, which were cryopreserved, from our collaborators, we regret that the potential experimental outcome from these mice may be only included in a future study due to time restriction.

*Reviewer #3: The updated version of the manuscript includes new set of data that answer several of my concerns. However, I have additional comments about the manuscript in its present form. 1) One major issue concerns the interpretation of the data obtained in vitro with BrdU incubation. According to their clarified BrdU experiment procedure. I noticed a misinterpretation of the data. Indeed, conversely to its fast clearance from brain tissue in vivo, the BrdU remains pretty stable (for several hours) in the culture medium and it is necessary to replace it after a short time to properly assess a single cohort of cells undergoing S-phase. This is an issue when it comes to monitor cell cycle exit rate of such cohort (Figure 2–Figure 6). In these bioassays, the cell cycle rate exit is totally biased by the continuous labelling (during 24 hours or so) of proliferative cells undergoing S-phase (which is indeed different between conditions). Same comments stand when assessing S-phase in proliferation assays with 6hours of incubation in BrdU-containing medium.*

We understand the reviewer’s concern that the length of BrdU exposure may bias our results and interpretation. Therefore, we thoroughly tested various conditions before we used fixed length of BrdU incorporation for different assays (6 hours for the proliferation and 24 hours for the cell cycle exit assays).

First, we tested the incorporation rate of BrdU as well as cell cycle exit rate at multiple time points in native NPCs before electroporation. Given that miR-128 upregulation and downregulation would produce opposing cellular effects, we determined the baseline condition that would be able to clearly demonstrate changes, which was set at around 50% BrdU^+^/Dapi^+^ for the cell proliferation assay upon 6 hour after BrdU exposure. We also set the baseline around 50% BrdU^+^Ki67^-^/Dapi^+^ for the cell cycle exit assay which was reached upon 24 hours of BrdU exposure.

Furthermore, we have included additional data at multiple time points from the cell proliferation assay and cell cycle exit assay following electroporation with miR-Zip-128 constructs. We have found that the effects of miR-128 knockdown – increased cell proliferation compared to control and decreased cell cycle exit compared to control – could be observed at most of the time points, indicating that the BrdU exposure lengths used in the original manuscript did not dictate or biasexperimental outcomes. We included this data in Figure 2—figure supplement 6, and described these results in the revised manuscript (subsection “miR-128 regulates the proliferation and differentiation of NPCs in vitro”, last paragraph).

2) The conclusion about mitotic spindle orientation (lanes 169-17) is incorrect. The Figure 3 indeed show that acute modulation of miR128 expression affect spindle orientation. It is not possible to conclude that default of mitotic spindle orientation leads to impaired APs division (symmetric vs asymmetric). This has been discussed in several recent publications. A clonal analysis would be required to follow the fate of the AP progenies after miR-128 modulation. One complementary set of experiments to decipher the functional impact of mitotic spindle orientation modification would be to focally electroporate APs (with miR-128 or miR-Zip-128) and identify the fate of the progenies 24 hours later (Tbr1 for neuron, Pax6 for APs and Tbr2 for IPs).

We agree with the reviewer that one cannot conclude the fate of AP progeny solely based on changes in mitotic spindle orientation alone and that is precisely the reason why we had followed up the spindle orientation findings with experiments to identify the fate of AP progenies. In our original manuscript, we found that miR-128 manipulation affected the balance between AP (PAX6 and SOX2 positive cells) and IP (TBR2 positive cells) generation (Figure 3).

We also acknowledge that recent studies have found that spindle orientation might not be the only mechanism responsible for fate determination in APs (Konno et al., 2008; Noctor et al., 2008; Peyre and Morin, 2012) and did not exclude the involvement of other potential mechanisms. We added these points in our revised manuscript (subsection “miR-128 promotes neuronal differentiation in the developing cortex in vivo”).

For the proposed clonal analysis, given that more than 85% of electroporated, GFP^+^ APs could be labelled with EdU upon 24 hours of exposure (data not shown), we could use GFP signal as a surrogate marker for EdU, that made EdU or BrdU incorporation assay redundant, when we perform analysis using PAX2 and TBR2 immunostaining. In addition, when APs were labelled with EdU for 24 hours immediately following electroporation for traditional clonal analysis, we observed that the knockdown of miR-128 increases Pax6-positive cells, but decreases Tbr2-positive cells in the EdU-positive cell population (Figure 10) in a manner that follows very closely with our previous findings in our first manuscript (original manuscript Figure 3). Taken together, the APs focally electroporated with miR-128 knockdown constructs increased APs, but decreased BPs.

However, we failed to observe cells that were double positive for both Tbr1 and EdU24 hours after EdU labelling (Figure 10), indicating there was negligible direct neurogenesis from the electroporated APs.

Author response image 3.Clonal analysis reveals the fate of progeny derived from electroporated progenitor cells.In utero electroporation was performed on E13.5 embryos to introduce either miR-128-ZIP or a control construct into NPCs lining the ependymal layer. Ependymal NPCs were further labeled with EdU immediately following electroporation. (**A-C**) At E14.5, tissue sections were acquired for EdU detection and immunostained with antibodies against PAX6 (**A**) and TBR2 (not shown). (**B**) Quantification of PAX6- EdU- GFP-triple-positive cells over EdU- GFP-double-positive cells shows significant increase of *Pax6* positive cells in progeny from miR-128-ZIP electroporated NPCs compared to control. (**C**) Quantification of TBR2- EdU- GFP-triple-positive cells over EdU- GFP-double-positive cells shows significant decrease of TBR2-positive cells in progeny from miR-128-ZIP electroporated NPCs compared to control. (**D**) Tissue sections acquired at E14.5, and processed for EdU detection and immunostained with antibodies against TBR1revealed minimal overlap between EdU-positive cells and TBR1-positive cells. In (**A**), circles show GFP- EdU-double-positive cells, blue circles show PAX6-negative cells among GFP-EdU- double-positive cells. Scale bar, 10 um. (**D**) Scale bar, 100 um. More than 500 GFP-positive cells were counted for each condition. At least three sets of independent experiments were performed. The values represent the means ± s.d. (n=3). Student’s *t*-test, differences were considered significant at ****P<0.001.***DOI:**
http://dx.doi.org/10.7554/eLife.11324.037

[Editors' note: further revisions were requested prior to acceptance, as described below.]

*The current version of the manuscript with additional data has significantly improved, compared with the previous version.*

*However, there is still one remaining point that should be fixed. Like mentioned previously, keeping BrdU in NPC culture (which is more stable than in vivo) to assess proliferation is OK if the authors do not refer to a single cohort of cells that have undergone S-phase (otherwise the authors should have only treated cells for about an hour with BrdU).*

*Now, concerning the cell cycle exit rate, we usually monitor a single cohort of cell that has exited the cell cycle. It is thus not possible to measure it properly if the BrdU is kept during 24 hours in vitro (stable) because several cohorts of cells will undergo S-phase at different timing, further leading to biased analyses. However, the cell cycle exit is properly addressed in vivo (where clearance of BrdU is fast). I would thus suggest to get rid of the in vitro data that are poorly executed.*

We agree with the reviewers and have made corrections in the revised manuscript. First, we have added statements (subsection “miR-128 regulates the proliferation and differentiation of NPCs in vitro”) to clarify that our in vitro proliferation assay was done without synchronizing the cell cycle of the NPCs and thus the change in proliferation rate may not represent that of a single cohort, but rather of a heterogeneous group from all phases of the cell cycle (Bez et al., 2003). In addition, as suggested by the reviewers, we have removed the in vitro cell cycle exit data from the manuscript.